# CMIP6 Multi-model Assessment of Northeast Atlantic and German Bight Storm Activity

Daniel Krieger[1,2,3] and Ralf Weisse[3]

[1]Max Planck Institute for Meteorology, Hamburg, Germany
[2]Institute of Oceanography, Universität Hamburg, Hamburg, Germany
[3]Institute of Coastal Systems – Analysis and Modeling, Helmholtz-Zentrum Hereon, Geesthacht, Germany

**Correspondence:** Ralf Weisse (ralf.weisse@hereon.de)

**Abstract**

We assess the evolution of Northeast Atlantic and German Bight storm activity using both model simulations and observational data. Our analysis includes the CMIP6 multi-model ensemble and the Max Planck Institute Grand Ensemble (MPI-GE) under CMIP6 forcing, evaluated across historical forcing and three future emission scenarios. Storm activity is quantified via upper percentiles of geostrophic wind speeds, derived from horizontal gradients of mean sea-level pressure. Observational datasets are employed to benchmark and validate the modeled storm characteristics, enhancing the robustness of our assessment. We detect robust downward trends for Northeast Atlantic storm activity in all scenarios, and weaker but still downward trends for German Bight storm activity. In both the multi-model ensemble and the MPI-GE, we find a projected increase in the frequency of westerly winds over the Northeast Atlantic and northwestesrly winds over the German Bight, and a decrease in the frequency of easterly and southerly winds over the respective regions. We also show that despite the projected increase in the frequency of wind directions associated with increased cyclonic activity, the 95th percentiles of wind speeds from these directions decrease, leading to lower overall storm activity. Lastly, we detect that the change in wind speeds strongly depends on the region and percentile considered, and that the most extreme storms (> 99th percentile) may become stronger or more likely in the German Bight in a future climate despite reduced overall storm activity.

## 1 Introduction

Strong winds and intense precipitation associated with extra-tropical cyclones pose significant weather-related hazards across the mid-latitudes of the Northern Hemisphere. Individually, these phenomena can result in severe wind damage to buildings and infrastructure (e.g., Heneka and Ruck, 2008), as well as inland (e.g., Luca et al., 2017) and coastal flooding (e.g., Wadey et al., 2015). When occurring simultaneously, they may trigger compound flooding events, such as the joint occurrence of elevated river discharge and storm surges (e.g., Heinrich et al., 2023), or the combination of heavy local precipitation and storm surges that inhibit drainage in coastal lowlands (e.g., Bormann et al., 2024).

Many coastal impacts are highly sensitive to the direction of approaching weather systems. Storm surge height, for instance, is strongly influenced by wind direction and its alignment with coastal geometry (e.g., Ganske et al., 2018). Wave-related hazards are particularly dependent on fetch length, which is inherently direction-dependent (e.g., Schmager et al., 2008), and wave direction itself plays a critical role in determining the extent and location of coastal erosion (e.g., Soomere and Viška, 2014). These directional dependencies must be considered when assessing cyclone-related risks in coastal regions.

In the Northern Hemisphere, there are two regions where extra-tropical cyclones statistically occur most frequently, the North Pacific and the North Atlantic (e.g., Shaw et al., 2016). These regions are commonly referred to as storm tracks (e.g., Blackmon et al., 1977; Shaw et al., 2016). In the following we focus on storms and storm tracks over the North Atlantic.

Because of their negative impacts on society, possible future changes of storms over the North Atlantic as a consequence of anthropogenic climate change have gained considerable attention in recent years. A comprehensive literature review was provided by Feser et al. (2015). Reviewing the results from 50 publications they found that about half of the studies concluded an increase in the number of storms by the end of the 21st century while the other half reported decreasing trends. Most studies that indicated an increase in storm numbers covered the North Atlantic north of $60°$ N. For the North Atlantic south of $60°$ N, more studies projected a decrease in storm numbers.

Many pre-CMIP3 and CMIP3 (Meehl et al., 2007) studies reported a poleward shift of the North Atlantic storm track (e.g., Fischer-Bruns et al., 2005; Bengtsson et al., 2009) while newer studies using data from the CMIP3/CMIP5 database emphasized an eastward extension of the North Atlantic winter storm track instead (e.g., Ulbrich et al., 2008; Zappa et al., 2013). Based on the results of analyses of the CMIP5 simulations, the IPCC's 5th assessment report (Kirtman et al., 2013) concluded that the number of extra-tropical cyclones composing the storm tracks is projected to weakly decline in the order of a few percent by 2100. At the same time, a reduction in the number of extra-tropical cyclones with very high surface winds, both in the extended winter season (Chang, 2018) and annually (Seiler and Zwiers, 2016) was reported as a robust signal in CMIP5 simulations (Lee et al., 2021).

In the IPCC's 6th assessment report and based on the analyses of 13 models from the CMIP6 ensemble (Eyring et al., 2016), it was concluded that there is overall low agreement among models regarding changes in extra-tropical cyclone density in the North Atlantic during boreal winter (Lee et al., 2021). This low model agreement reflects considerable uncertainty in the future evolution of storm tracks, both in their density and geographical location. Because local wind speed extremes are closely linked to both the intensity and the position of storm tracks, such uncertainty translates directly into a high degree of uncertainty regarding the future occurrence and distribution of extreme wind events at specific locations within the North Atlantic sector (e.g., Zappa et al., 2013; Barcikowska et al., 2018).

Priestley and Catto (2022) analyzed future changes in the extratropical storm tracks and cyclone intensity in an ensemble of nine CMIP6 simulations from which the necessary data for the analyses were available. They found that in the three emission scenarios SSP2-4.5, SSP3-7.0, and SSP5-8.5 the total number of cyclones over the North Atlantic decreased in the order of 5-7% by 2100 with the stronger decreases detected in the higher emission scenarios in both winter and summer seasons. At the same time, an increase in the number of wintertime intense cyclones was reported. All scenarios showed a similar pattern of storm track change. In the North Atlantic along the Greenwich Meridian, Priestley and Catto (2022) reported a tripolar pattern

of change with an increase in the track density over the British Isles and a decrease over the subtropical central North Atlantic and the Norwegian Sea.

Harvey et al. (2020) compared the response of the Northern Hemisphere storm tracks to climate change in the CMIP3, CMIP5, and CMIP6 climate models. Comparing historical simulations with the SRES-A1B simulations from CMIP3, the RCP4.5 simulations from CMIP5, and the SSP2-4.5 simulations from CMIP6, they concluded that the spatial patterns of the climate change response of the North Atlantic storm track remain similar in the CMIP3, CMIP5, and CMIP6 models. Using 19 models from CMIP3, 38 from CMIP5, and 14 from CMIP6, Harvey et al. (2020) further concluded that for the North Atlantic, the main response of the models is strengthening and an extension of the winter storm track that is most pronounced in the CMIP3 and CMIP6 models. The pattern described reveals the same spatial structure as reported by Priestley and Catto (2022) for nine models from the CMIP6 simulations.

Numerous metrics were used in the literature to quantify changes in storm activity (e.g., Yau and Chang, 2020). Metrics that correlate well with the impacts of extra-tropical cyclones are, for example, changes in local upper percentiles of near-surface wind speeds (e.g., Alexandersson et al., 1998; Paciorek et al., 2002) since buildings and infrastructures are generally designed according to the local climatological wind conditions. Schmidt and von Storch (1993) developed a proxy in which upper percentiles of geostrophic wind speeds are derived from triangles of atmospheric pressure observations. Krueger and von Storch (2011) have shown that variations in the statistics of strong geostrophic wind speeds well describe the variations of statistics of near-surface wind speeds. Although the proxy was originally developed to address the lack of homogeneity in time series of wind speed measurements (e.g., The Wasa Group, 1998; Alexandersson et al., 1998), it has been widely used to address changes in observed (e.g., Alexandersson et al., 2000; Paciorek et al., 2002; Matulla et al., 2008; Krueger et al., 2019; Krieger et al., 2021) or model-based (reanalysis) time series (e.g., Wang et al., 2009, 2011; Krueger et al., 2013). An advantage of the geostrophic proxy over the analysis of actual wind speeds in model data is the independence of geostrophic wind speeds on surface wind parametrizations, which may differ between models and induce biases in the analysis of absolute wind speeds and their trends.

A central challenge is that most existing studies are limited by model selection, diagnostic constraints, or incomplete sampling of plausible climate outcomes. Many rely on a restricted subset of CMIP models due to data availability, and often focus either on mean trends or a narrow set of extreme metrics. Meanwhile, the role of stochastic climate "noise" and the full envelope of possible outcomes, including the change in extreme events that may not be captured by analyzing means or quartiles, but are crucial for robust risk assessment, are only partially addressed by traditional multi-model ensembles. Large parts of decision making in the coastal protection sector rely on these estimates of variability, uncertainty, and the future change of event distributions which multi-model ensembles like the CMIP6 suite in itself are less suited to provide (Paté-Cornell, 1996; Weaver et al., 2013).

To address these gaps, our study provides a more comprehensive assessment of projected storm activity changes by leveraging two methodological advances. First, we employ the pressure-based proxy developed by Schmidt and von Storch (1993) as it allows us to consider a larger ensemble of 32 CMIP6 models that allows a more comprehensive assessment of changing Northeast Atlantic storm activity under different anthropogenic forcing scenarios: SSP1.2-6, SSP2-4.5, and SSP5-8.5. Sec-

ond, we complement the multi-model ensemble with the 50-member Max Planck Institute Grand Ensemble (MPI-GE) under CMIP6 forcing (Olonscheck et al., 2023), a single-model initial condition large ensemble (SMILE). The MPI-GE with its high-resolution output allows us not only to illustrate the range of outcomes associated with different initial climate states under identical external forcing, but also to explore the robustness, variability, and physical plausibility of projected changes in storm activity, including at the most extreme percentiles. Rather than focusing solely on internal variability, we use the SMILE to map the spectrum of physically consistent futures, highlight tail risks, and test the sensitivity of our findings to initial conditions—an essential consideration for decision support and adaptation planning (e.g., Mankin et al., 2020; Weaver et al., 2013). With these two advances, we aim at answering the following research questions:

– How robust and consistent are projected changes in storm activity and wind direction across an expanded set of CMIP6 models, when assessed using a pressure-based proxy?

– How do these multi-model forced responses compare to the spread of plausible outcomes provided by the high-frequency MPI-GE, and how does the MPI-GE project changes in frequency and characteristics of the most extreme wind events?

The manuscript is structured as follows: In Section 2, we introduce the datasets, methods, and regions used in this study. Section 3.1 estimates the forced response of German Bight and Northeast Atlantic storm activity and wind direction distributions to anthropogenic climate change in the CMIP6 multi-model ensemble. Section 3.2 follows up with comparison of storm activity in the MPI-GE with the multi-model ensemble, as well as an estimate of the future risk of very extreme events by comparing changes in absolute geostrophic wind speed distributions. Section 4 discusses our findings and provides a short outlook, while concluding remarks are given in Section 5.

## 2 Methods and Data

### 2.1 Data

In this study, we employ climate model output from the sixth phase of the Coupled Model Intercomparison Project (CMIP6; Eyring et al., 2016). We use mean sea-level pressure (MSLP) data from historical simulations spanning the time period 1850-2014, as well as future scenario simulations under SSP1.2-6, SSP2-4.5, and SSP5-8.5 forcings, each spanning the time period 2015-2100. We constrain our analysis to those CMIP6 models for which MSLP data from the historical and the three afore-mentioned scenario simulations is available at daily resolution (Table 1). Additionally, we examine the 50-member CMIP6 version of the Max Planck Institute Earth System Model (MPI-ESM-LR) at three-hourly resolution, which we refer to as the Max Planck Institute Grand Ensemble (MPI-GE; Olonscheck et al., 2023). While the three-hourly output of MPI-ESM-LR, i.e., the MPI-GE, is not included in the CMIP6 multi-model analysis, the regular daily output of MPI-ESM-LR is included as one of 32 models. Throughout this manuscript, MPI-GE always refers to the separately analyzed three-hourly dataset produced with MPI-ESM-LR.

**Table 1.** List of the 32 CMIP6 models used in this study and their ensemble sizes.

| Model | Number of Ensemble Members | | | | Reference |
|---|---|---|---|---|---|
| | Historical | SSP1-2.6 | SSP2-4.5 | SSP5-8.5 | |
| ACCESS-CM2 | 1 | 3 | 3 | 3 | Bi et al. (2020) |
| ACCESS-ESM1-5 | 1 | 1 | 3 | 1 | Ziehn et al. (2020) |
| BCC-CSM2-MR | 2 | 1 | 1 | 1 | Wu et al. (2019) |
| CESM2 | 11 | 1 | 1 | 3 | Danabasoglu et al. (2020) |
| CESM2-WACCM | 3 | 1 | 5 | 5 | Danabasoglu et al. (2020) |
| CMCC-CM2-SR5 | 1 | 1 | 1 | 1 | Cherchi et al. (2019) |
| CMCC-ESM2 | 1 | 1 | 1 | 1 | Cherchi et al. (2019) |
| CNRM-CM6-1 | 20 | 6 | 6 | 6 | Voldoire et al. (2019) |
| CNRM-CM6-1-HR | 1 | 1 | 1 | 1 | Voldoire et al. (2019) |
| CNRM-ESM2-1 | 10 | 5 | 3 | 5 | Séférian et al. (2019) |
| CanESM5 | 18 | 50 | 20 | 20 | Swart et al. (2019) |
| EC-Earth3 | 73 | 7 | 7 | 8 | Döscher et al. (2022) |
| EC-Earth3-Veg | 3 | 3 | 3 | 3 | Döscher et al. (2022) |
| EC-Earth3-Veg-LR | 1 | 1 | 1 | 1 | Döscher et al. (2022) |
| FGOALS-g3 | 2 | 1 | 1 | 1 | Li et al. (2020) |
| GFDL-ESM4 | 1 | 1 | 1 | 1 | Dunne et al. (2020) |
| HadGEM3-GC31-LL | 4 | 1 | 4 | 1 | Kuhlbrodt et al. (2018) |
| IITM-ESM | 1 | 1 | 1 | 1 | Swapna et al. (2018) |
| INM-CM4-8 | 1 | 1 | 1 | 1 | Volodin et al. (2018) |
| INM-CM5-0 | 10 | 1 | 1 | 1 | Volodin et al. (2017) |
| IPSL-CM6A-LR | 31 | 6 | 3 | 3 | Boucher et al. (2020) |
| KACE-1-0-G | 1 | 3 | 1 | 3 | Lee et al. (2020) |
| KIOST-ESM | 1 | 1 | 1 | 1 | Pak et al. (2021) |
| MIROC-ES2L | 1 | 3 | 1 | 1 | Hajima et al. (2020) |
| MIROC6 | 34 | 3 | 3 | 3 | Tatebe et al. (2019) |
| MPI-ESM1-2-HR | 10 | 2 | 2 | 2 | Müller et al. (2018) |
| MPI-ESM1-2-LR | 50 | 50 | 50 | 50 | Mauritsen et al. (2019) |
| MRI-ESM2-0 | 7 | 1 | 1 | 2 | Yukimoto et al. (2019) |
| NESM3 | 1 | 2 | 2 | 2 | Cao et al. (2018) |
| NorESM2-LM | 1 | 1 | 3 | 1 | Seland et al. (2020) |
| NorESM2-MM | 1 | 1 | 1 | 1 | Seland et al. (2020) |
| UKESM1-0-LL | 8 | 5 | 6 | 5 | Sellar et al. (2019) |

## 2.2 Target Regions

We focus our analysis on two regions of the North Atlantic storm track, namely the large-scale Northeast Atlantic Ocean and the smaller-scale German Bight. For the Northeast Atlantic Ocean, we calculate storm activity for a set of ten triangles mimicking those used in Krueger et al. (2019). The German Bight is represented by a triangle with the cornerpoints List/Sylt, Norderney, and Hamburg-Fuhlsbüttel (Fig. 1, Tables 2, 3).

As both the Northeast Atlantic Ocean and North German Plain triangles are originally based on observation sites which may not be located near a model gridpoint, we ensure that we approximate the triangles by choosing those gridpoints in each respective model that lie closest to the original observation site.

However, in regions with complex orography, particularly for sites such as Bodø and Bergen, this approach may introduce some distortions. For instance, the nearest grid point in a given model may lie inland or at a different elevation than the observational site, whereas in another model the nearest grid point may be located over flatter terrain or the ocean. Although we use MSLP rather than surface pressure, such differences in grid point selection can lead to small inconsistencies across models with unequal pressure reduction algorithms, especially in areas with steep topography. We therefore acknowledge that this limitation may slightly affect the comparability of storm activity estimates for these specific locations, which is a common problem among all studies that use pressure-based proxies.

To minimize further methodological issues, we ensure that the three selected grid points do not fall on a straight line (e.g., by sharing the same latitude or longitude), which would otherwise preclude a meaningful geostrophic wind calculation due to an enclosed area of zero. In such cases, we slightly adjust the position of one grid point to form a proper triangle. Specifically, we move the grid point corresponding to the observation site that is geometrically furthest from the initially assigned grid point. This adjustment is limited to a single grid cell in the nearest orthogonal direction to preserve the original geometry as closely as possible while ensuring a valid triangle. Finally, we note that all pressure gradient and geostrophic wind calculations are based on the selected model grid points, rather than the exact locations of the original observation sites. While these choices are standard in pressure-based storm activity proxies, we recommend that future studies in highly orographically complex regions consider sensitivity tests or more advanced interpolation methods to further reduce potential bias.

## 2.3 Calculation of Storm Activity

The calculation of storm activity follows the approach of Schmidt and von Storch (1993) and Alexandersson et al. (1998). We define storm activity as annual 95th percentiles of geostrophic wind speeds, which we derive from triplets of simultaneous three-hourly-mean (MPI-GE) or daily-mean (full CMIP6 suite) MSLP data. The annual percentiles are standardized member-wise by subtracting the 1961-1990 mean and dividing by the 1961-1990 standard deviation of the respective member. The standardization reference period of 1961-1990 follows both Krueger et al. (2019) and Krieger et al. (2021). For the Northeast Atlantic Ocean, we standardize the time series individually for each triangle and then average over the ten standardized time series, again separately for each ensemble member. To compare the modeled storm activity to observations, we include time series of observed storm activity from the Northeast Atlantic (Krueger et al., 2019) and the German Bight (Krieger et al., 2021)

in our analysis. Observed storm activity is calculated similarly to the modeled counterpart, evaluating annual 95th percentiles of geostrophic wind speeds, derived from MSLP measurements at the locations listed in Tables 2 and 3. The observed time series cover the periods of 1897-2019 (German Bight) and 1875-2016 (Northeast Atlantic). Data sources and a more detailed description of resolution and quality control are found in the respective studies. In addition to annual storm activity, we also calculate the annual distributions of the geostrophic wind direction, segmented into the 16 main cardinal directions.

The CMIP6 model suite used in this study consists of multiple model ensembles, the sizes of which depend on the model and the scenario. To avoid overweighting larger ensembles in this multi-model analysis, we use a bootstrapping approach and repeatedly select one random ensemble member from each model with replacement. We repeat the bootstrapping 1000 times and define the mean over the resulting 1000 sets of 32 model simulations from 32 different climate models as our CMIP6 multi-model mean. We perform this bootstrapping separately for the historical runs and each scenario, as ensemble sizes vary between scenarios.

## 2.4 Estimating Statistical Significance

To estimate whether changes between the historical reference period and the three end-of-century climates are statistically significant, we employ a bootstrapping approach (Efron and Tibshirani, 1986). From each 30-year period, we draw 1000 random samples with replacement, each one with the size of the original sample. From the pairs of randomly drawn samples, we calculate the distribution of possible differences between historical and future climates, and define the 0.025- and 0.975- quantiles of differences as the boundaries of the 95%-confidence interval. Should the confidence interval exclude zero, we reject the null hypothesis that the changes are not significant.

**Table 2.** Coordinates of the locations used for storm activity calculation.

| Gridpoint | Latitude (°N) | Longitude (°E) |
|---|---|---|
| **Northeast Atlantic** | | |
| Jan Mayen (J) | 70.93 | -8.67 |
| Bodø (O) | 67.27 | 14.43 |
| Bergen (B) | 60.38 | 5.33 |
| Aberdeen (A) | 57.20 | -2.20 |
| Valentia (V) | 51.93 | -10.25 |
| Stykkisholmur (S) | 65.08 | -22.73 |
| Torshavn (T) | 62.02 | -6.77 |
| de Bilt (D) | 52.10 | 5.18 |
| Vestervig (G) | 56.73 | 8.27 |
| Nordby (N) | 55.47 | 8.48 |
| **North Germany** | | |
| List | 55.01 | 8.41 |
| Norderney | 53.71 | 7.15 |
| Hamburg-Fuhlsbüttel | 53.63 | 9.99 |

**Table 3.** List of triangles and their gridpoints.

| Triangle | Gridpoint 1 | Gridpoint 2 | Gridpoint 3 |
|---|---|---|---|
| TSO | Torshavn | Stykkisholmur | Bodø |
| BTA | Bergen | Torshavn | Aberdeen |
| TOB | Torshavn | Bodø | Bergen |
| AVT | Aberdeen | Valentia | Torshavn |
| BGA | Bergen | Vestervig | Aberdeen |
| AVD | Aberdeen | Valentia | de Bilt |
| AGD | Aberdeen | Vestervig | de Bilt |
| VST | Valentia | Stykkisholmur | Torshavn |
| JSO | Jan Mayen | Stykkisholmur | Bodø |
| TNB | Torshavn | Nordby | Bergen |
| German Bight | List | Norderney | Hamburg-Fuhlsbüttel |

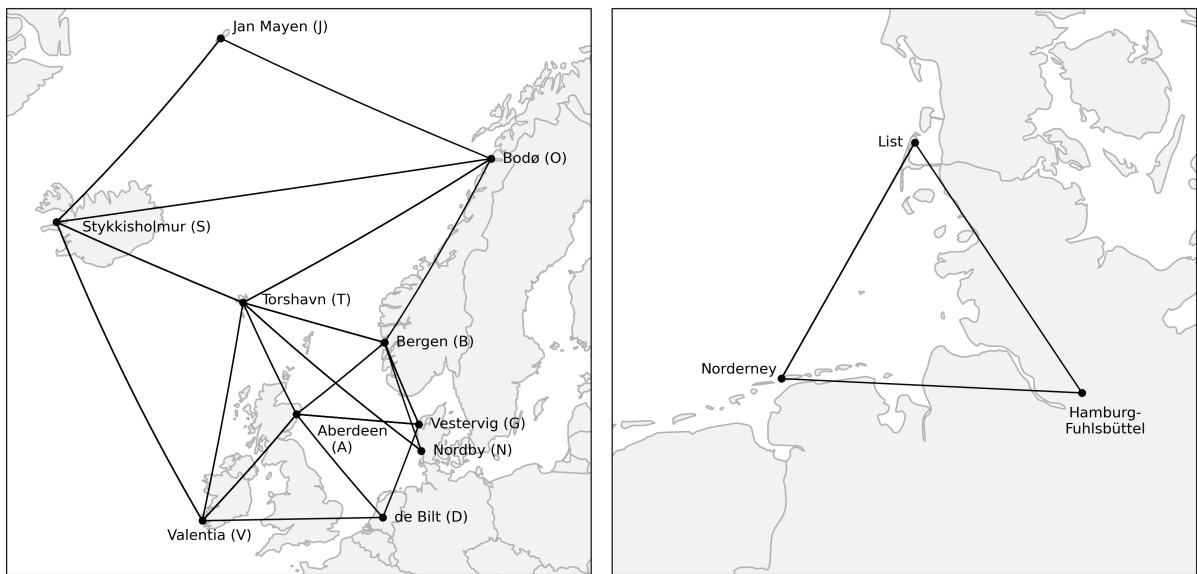

**Figure 1.** Maps of the Northeast Atlantic (left) and German Bight (right) stations and triangles.

## 3 Results

### 3.1 Forced Response - A Multi-Model View

**Storm Activity**

We first analyze the projected evolution of Northeast Atlantic storm activity (NeASA) and German Bight storm activity (GBSA) in the full CMIP6 multi-model suite. The results of the multi-model analysis are an indicator of the forced response of the climate system, and in particular storm activity, to the projected changes in greenhouse gas forcing.

In the historical period, the multi-model mean shows interdecadal fluctuations in NeASA, with a slight downward trend over time and a slight downward trend (Figs. 2a and 3c). The trend computed from bootstrapped medians across the historical period is weaker than the observed trend, while the range of trends computed from all members includes the observed trend. The interdecadal variations of storm activity likely reflect the response to external forcing as represented across models, rather than internally generated oscillations. Under the three considered future scenarios (SSP1-2.6, 2-4.5, and 5-8.5), NeASA is projected to decrease to approximately 0.5-0.7 standard deviations below that of the reference timeframe, with most members showing a negative trend throughout the projection period (Fig. 3c). While the projected decrease of NeASA is observed under all three greenhouse gas forcing scenarios, the bootstrapped median trends are strongest in the high-emission SSP5-8.5 scenario (Fig. 3a), indicating a inverse relationship between projected storm activity and global warming in the CMIP6 suite. In all three scenarios, none of the bootstrapped multi-model ensembles suggests an end-of-century (EoC, 2071-2100) storm activity above that of the historical reference period from 2050 onward.

Notably, the bootstrapped uncertainty range is much smaller than the variability of observed NeASA throughout the historical periods, likely caused by the calculation of the multi-model mean which always includes the same member from those models with an ensemble size of 1. Thus, the bootstrapped multi-model means are always nudged towards the mean of these 16 models, restricting the generation of uncertainty to the remaining 16 models. When the selection of members is limited to those models with an ensemble size of at least 5 members (Fig. 2c), the uncertainty in the forced response increases, as contributions from each model vary between bootstraps. The uncertainty resulting from selecting only members from larger ensembles is much closer to the observed uncertainty than that resulting from bootstrapping all models. For the projections, fewer models with 5 or more ensemble members are available than for the historical period (compare Table 1). Consequently, the uncertainty in the projections increases even further than that of the historical period, leading to a small but non-zero fraction of bootstrapped multi-model means which show individual years with NeASA levels of above 0 in an EoC climate under all scenarios. Still, the 2071-2100 mean climate is robustly projected to drop below 0, following the evolution seen in Fig. 2a, and 100 % of all bootstraps agree on a 2071-2100 average NeASA below 0, irrespective of the forcing scenario. Taking all members from all models into consideration without bootstrapping or weighting, the observed time series of NeASA lies mostly within a band determined by $\pm$ one standard deviation around the mean, indicating that the full pool of ensemble members can represent the decadal variability present in the observations (Fig. 2e). While this is correct by definition for the reference period 1961-1990 as all timeseries are independently standardized with respect to this period, it also holds for the periods before and after the reference window.

Over the German Bight, the multi-model mean again displays interdecadal variability in storm activity (GBSA, Fig. 2b), however without any detectable long-term trend (Fig. 3d). While the observational record of GBSA (compare Fig. 2f) contains pronounced multidecadal variability, only weak indications of such features are evident in the ensemble mean, suggesting that these are not consistently reproduced by the externally forced response in the models. Contrary to the Northeast Atlantic, the projected change in GBSA follows much weaker trends (Fig. 3d) and all three scenarios depict a rather stationary evolution until the end of the century. Especially in the SSP2-4.5 scenario, the bootstrapped median trends are very close to 0, further suggesting stationarity (Fig. 3b). The bootstrapped multi-model means project a below-average GBSA with values of roughly 0.3-0.4 standard deviations below that of the reference period throughout most of the century. The GBSA in the high-emission SSP5-8.5 scenario lies slightly above that in the other two scenarios, so that any inverse relation between GBSA and global warming cannot be concluded from this analysis. Like in to NeASA projections, all bootstrapped multi-model means agree on the negative sign during the EoC climate in all three scenarios. Similar to the differences between the results of bootstrapping all models and bootstrapping only the models with an ensemble size of 5 or more for NeASA, the uncertainty is also increased for historical GBSA and even more for projected GBSA (Fig. 2d). Despite the large uncertainty ranges, the bootstrapped means (i.e., the thick lines in Fig. 2d) still agree on an EoC storm activity of below 0 in all scenarios. Similar to the historical period of NeASA, the pool of all members contains the observed time series of GBSA within its $\pm 1\sigma$ band (Fig. 2f).

**Wind Direction**

Changes in storm activity are caused by changes in the wind speed distribution, which oftentimes go hand in hand with changes in the distribution of wind directions. Thus, we analyze the projected changes in the occurrence frequencies of wind directions under different greenhouse gas forcings.

For the Northeast Atlantic, the CMIP6 suite projects an increase in the frequency of southwesterly, westerly, and northwesterly wind components in an EoC climate, as well as a decrease of the frequency of easterly and southerly winds (Fig. 4a). The magnitude of increase or decline follows the strength of the emissions, with the SSP5-8.5 scenario showing the largest changes. It is notable that those wind directions which are already favored in the historical period further increase in frequency. The directional changes are consistent for the German Bight, where the CMIP6 suite shows the biggest increases for northwesterly, northerly, and northeasterly winds, while simultaneously projecting decreases for the southeasterly and southerly components (Fig. 4b). In the SSP1-2.6 runs, decreasing frequencies for westerly winds can also be seen; these, however, change sign and are not statisticially significant anymore in the higher-emission SSP2-4.5 and SSP5-8.5 scenarios. Contrary to the Northeast Atlantic, the strongest frequency increases and decreases occur for those wind directions that occur rather infrequently, while the most common wind direction (west) shows almost no change until the end of the 21st century.

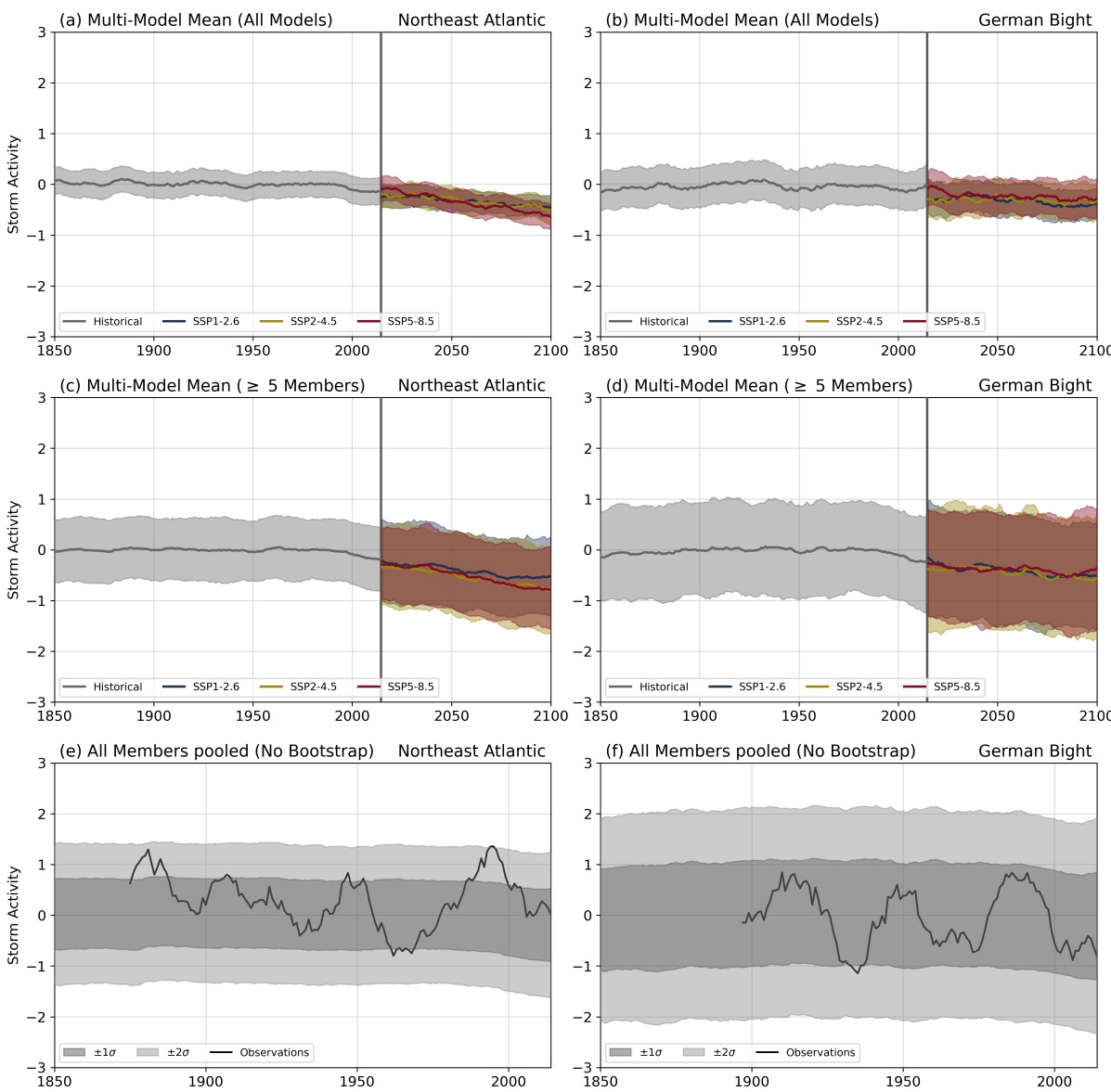

**Figure 2.** CMIP6 multi-model time series of (a,c,e) Northeast Atlantic and (b,d,f) German Bight storm activity for historic simulations (gray) and future scenarios (colors). Thick lines in (a)-(d) mark the multi-model mean, shaded areas indicate the range of the bootstrapped ensemble means. Bootstraps in (a) and (b) were taken from all models, bootstraps in (c) and (d) were taken from models with an ensemble size of at least 5 members for the respective scenario. Shadings in (e) and (f) show the range of 1 and 2 standard deviations of all pooled members for the historical period, with the observed storm activity added as a solid line. A 10-year moving average has been applied to all annual values.

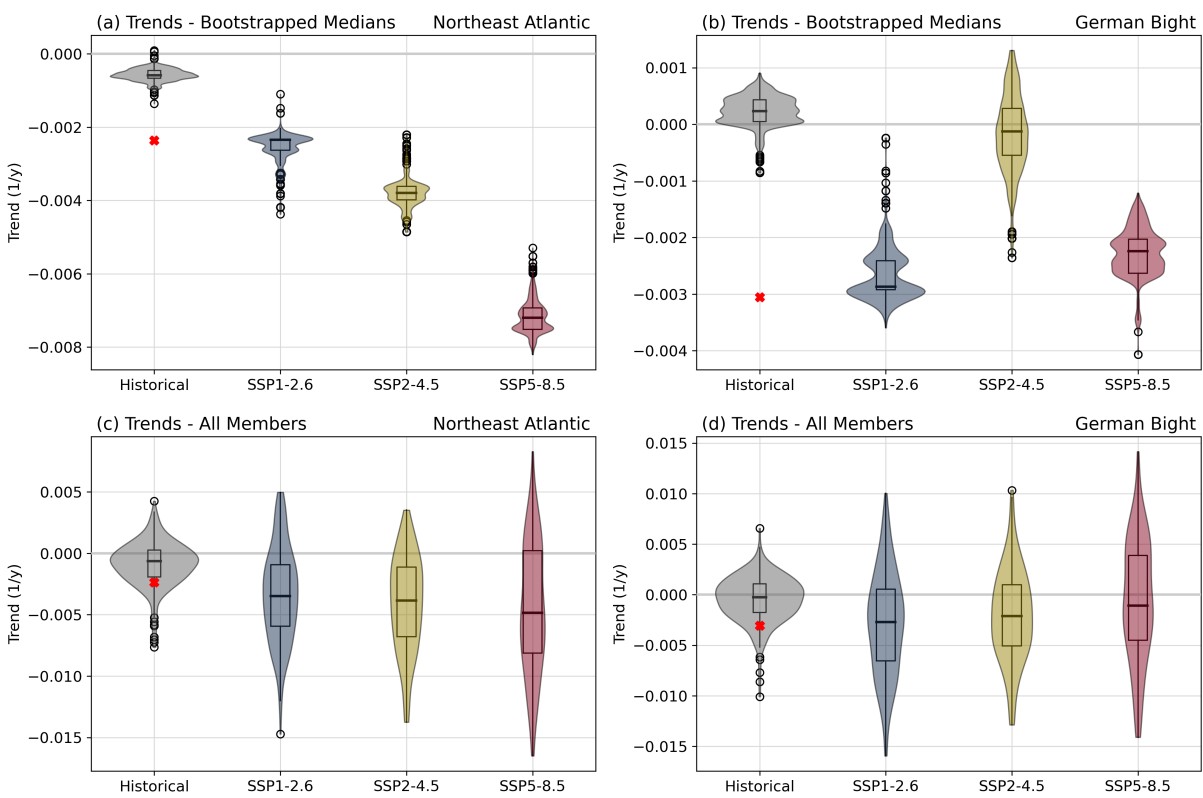

**Figure 3.** CMIP6 multi-model distributions of linear trends of (a,c) Northeast Atlantic and (b,d) German Bight storm activity for historic simulations (gray) and future scenarios (colors). (a) and (b) show the distributions of medians of 1000 bootstrapped sets, where one random member was drawn from each model. (c) and (d) display the distribution of trends from all members. Violins show the distributions of trends, box plots mark the median and interquartile range (IQR), with whiskers extending to 1.5 times the IQR. Red "x" markers in show the observed trends. Trends are computed over the entire available periods, i.e., 1850-2014 for historical runs, 2015-2100 for scenarios.

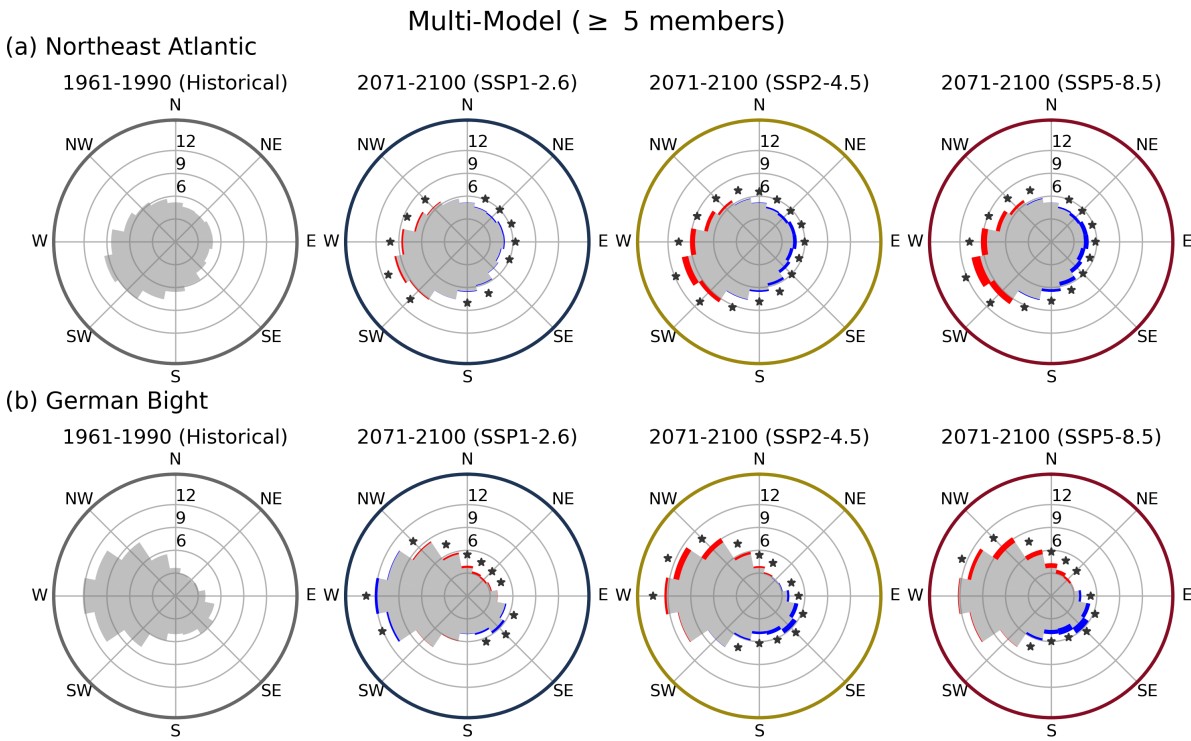

**Figure 4.** CMIP6 multi-model mean distributions of daily-mean (a) Northeast Atlantic and (b) German Bight wind directions for the historical period (1961-1990, left) and three end-of-century climates (2071-2100). Gray bars indicate the respective distributions of wind directions, red and blue colors highlight positive and negative changes between future and historical climates, respectively. Bootstraps only select from those models with 5 or more ensemble members for the respective scenario. Stars mark statistically significant changes ($p < 0.05$).

### 3.2 A SMILE approach with the high-frequency MPI-GE CMIP6

**Storm Activity and Wind Directions**

Understanding how the full distribution of storm activity—including not only mean values but also the extremes and directional shifts—responds to future climate forcing is crucial for impact assessments and risk management. Here, we use the 50-member Max Planck Institute Grand Ensemble (MPI-GE) to analyze projected changes across the storm intensity spectrum and investigate shifts in wind direction frequencies under different emissions scenarios.

While single-model initial-condition large ensembles (SMILEs) like the MPI-GE are powerful tools to disentangle externally forced climate signals from the envelope of possible realizations (internal variability), the primary focus of this section is on the response of different parts of the storm intensity and wind direction distributions to future climate change. Specifically, we examine how the extreme events in the tails of the wind speed distribution are projected to change and whether there are systematic changes in the occurrence frequencies of specific wind directions. To provide context, we first compare the ensemble mean evolution in MPI-GE to the CMIP6 multi-model trend, confirming that MPI-GE is representative of the forced response before focusing on distributional and extreme-event changes.

The historical simulations of the MPI-GE show a slighly above-average NeASA during the early period from 1850 to about 1930, followed by a gradual decline to near-normal states afterwards (Fig. 5a), yielding a modest negative trend in the historical period (Fig. 5e), close to the observed trend. Here, too, the ensemble mean displays weak multi-decadal variability, although this is less pronounced than in the observations. The spread among ensemble members ($\pm 1\sigma$) encompasses much of the observed historical variability (Fig. 5c), indicating that the range of outcomes simulated by the MPI-GE is consistent with past observed decadal variability. In all scenarios, the projected decline in NeASA is less pronounced in the MPI-GE than in the multi-model ensemble, with the ensemble mean stabilizing at about -0.3 to -0.4 standard deviations in the second half of the 21st century (Fig. 5a). For SSP5-8.5, the trend is weakest, reflecting low initial storm activity in the early scenario period. (Fig. 5e). Nevertheless, nearly all ensemble members agree on a below-average storm activity for end-of-century climates in all scenarios, with only rare exceptions in SSP5-8.5.

A similar pattern is found for GBSA: an initial increase in the late 19th/early 20th century, followed by a decline and weak trends across all scenarios (Figs. 5b, f). In all three projections, the MPI-GE shows an equilibrating behavior for most of the 21st century with a storm activity between -0.1 and -0.3 standard deviations. Similar to the CMIP6 projections, the high-emission SSP5-8.5 scenario shows the highest storm activity and remains above the other two scenarios, but the vast majority of ensemble members still point to below-average activity by the end of the century (74 % of members under SSP5-8.5 forcing, 92 % under SSP2-4.5, and 94 % under SSP1-2.6). Like for NeASA, the ensemble spread again captures most of the observed variability (Fig. 5d).

The MPI-GE mostly agrees with the CMIP6 suite on the directional changes over the Northeast Atlantic (Fig. 6a). Southwesterly to northwesterly directions are projected to increase, while northeasterly to southerly directions are projected to decrease, with the magnitude increasing with the level of emissions. For the German Bight, however, we observe some disparities between the MPI-GE and the CMIP6 suite. The strongest increases also include the westerly sector, but exclude the northeasterly

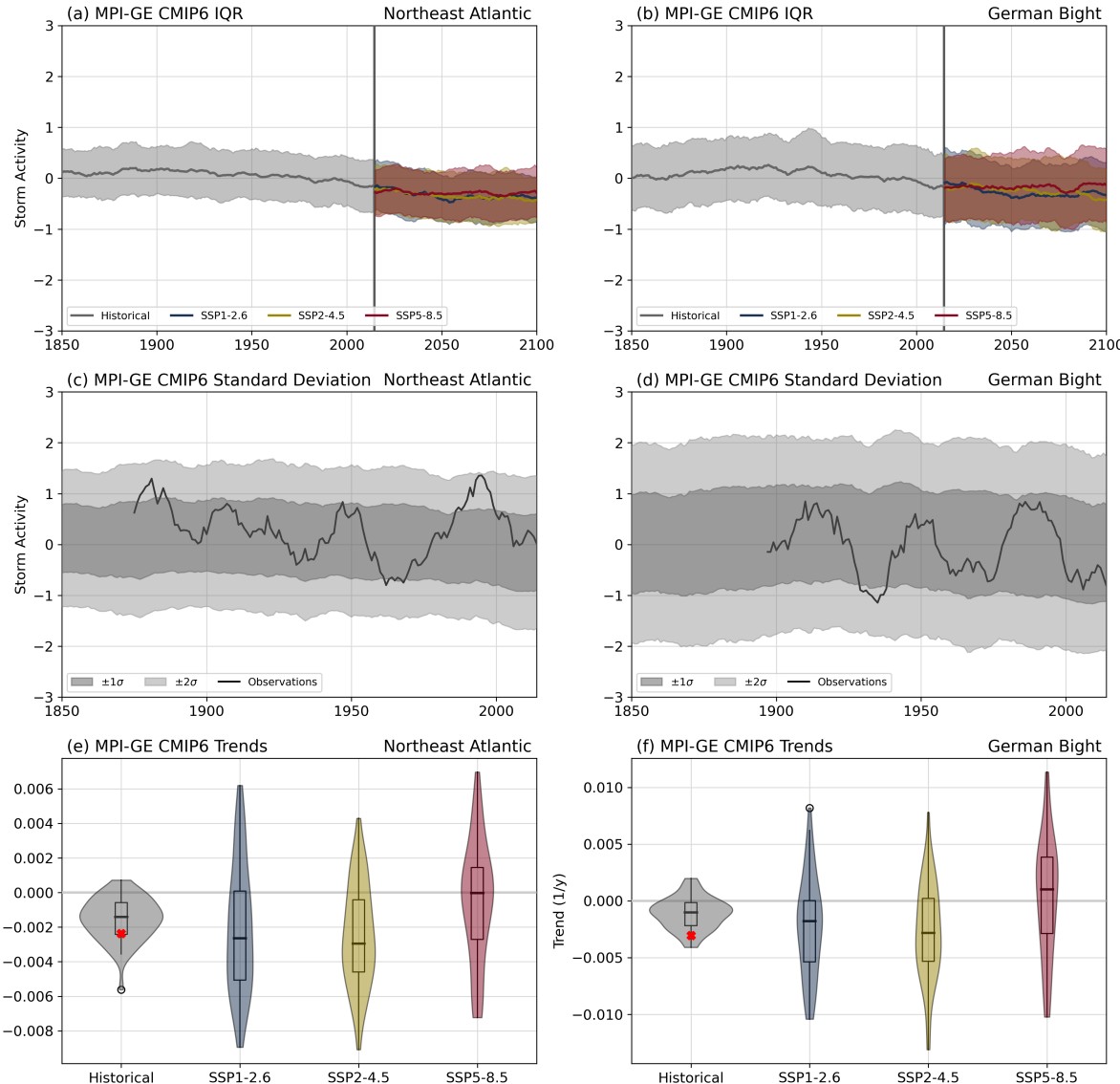

**Figure 5.** MPI-GE CMIP6 time series and linear trend distributions of (a,c,e) Northeast Atlantic and (b,d,f) German Bight storm activity for historic simulations (gray) and future scenarios (colors). Thick lines in (a) and (b) mark the ensemble mean, shaded areas indicate the interquartile range (IQR) of the 50-member ensemble. Shadings in (c) and (d) show the range of 1 and 2 standard deviations of all members for the historical period, with the observed storm activity added as a solid line. A 10-year moving average has been applied to all annual values in (a)-(d). Violins in (e) and (f) show the distributions of trends, box plots mark the median and interquartile range (IQR), with whiskers extending to 1.5 times the IQR. Red "x" markers in (e) and (f) show the observed trend.

270   directions. Overall, the pattern of frequency changes in the MPI-GE German Bight analysis (Fig. 6b) is rotated counterclockwise by about 45° compared to the CMIP6 multi-model counterpart (compare Fig. 4b). Furthermore, the general rule of larger

changes for higher-emission scenarios persists within the MPI-GE, whereas for the CMIP6 suite this is not entirely the case (compare SSP5-8.5 windroses in Figs. 4b and 6b).

Combining the findings for storm activity and wind direction, it appears counter-intuitive why the storm activity is projected to decrease even though the high-emission EoC climate may favor those wind directions that are typically associated with higher wind speeds and storms, i.e., southwesterly, westerly, and northwesterly. To disentangle this contradicting behavior, we analyze the projected changes of upper percentiles of absolute geostrophic wind speeds per cardinal direction and relate it to the changes in occurrence frequency in the MPI-GE. Here, we inspect absolute wind speeds which are not standardized and thus restrict this analysis to the single-model large-ensemble MPI-GE to avoid introducing inter-model biases. A comparison of direction-specific 95th percentiles between the SSP5-8.5 EoC climate and the historical reference in the German Bight (Fig. 7) shows that only southwesterly wind speeds are expected to increase in magnitude, while especially northwesterly winds may become significantly weaker in a future climate. Those cardinal directions for which higher 95th percentiles (SW) are expected simultaneously show a decrease in frequency, while more preferred directions in the future (NW) simultaneously weaken in intensity. As a result, the total storm activity, which is only based on the overall 95th percentiles and does not take direction into account, decreases in the EoC projections. Similar patterns can be found for most regions of the Northeast Atlantic, explaining the robust projected decrease in storm activity for NeASA as well (not shown).

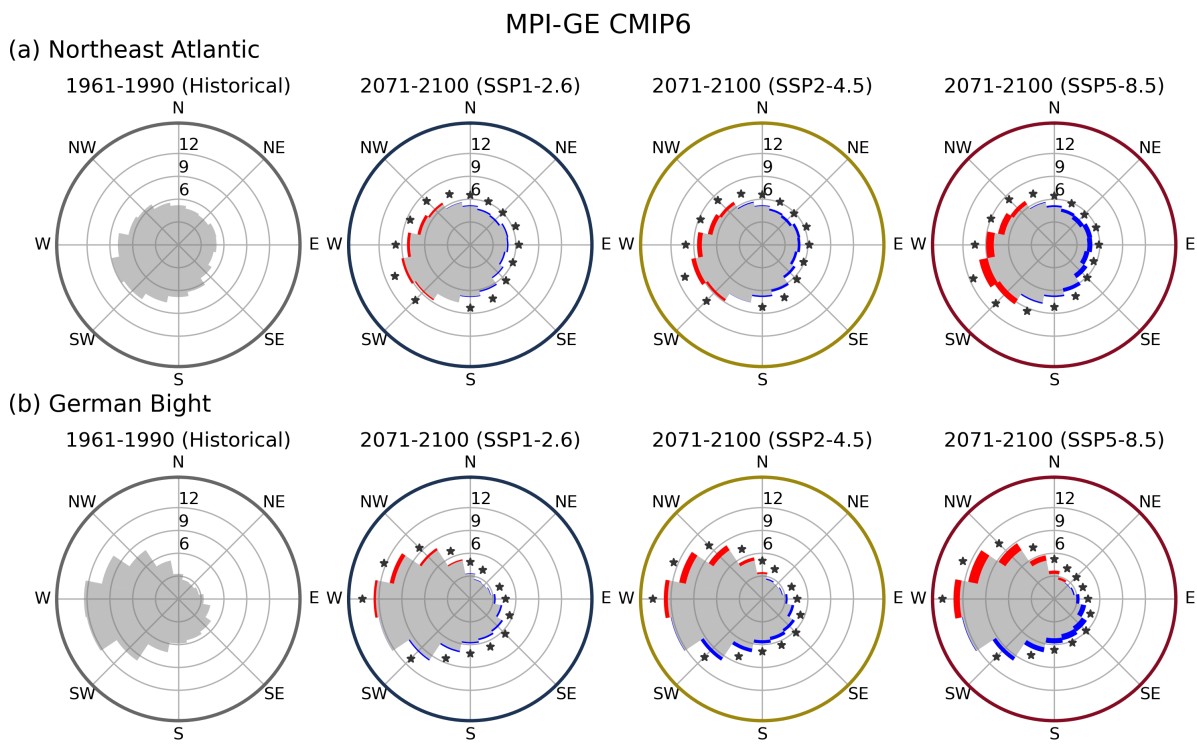

**Figure 6.** MPI-GE distributions of three-hourly (a) Northeast Atlantic and (b) German Bight wind directions for the historical period (1961-1990, left) and three end-of-century climates (2071-2100). Gray bars indicate the respective wind distribution, red and blue colors highlight positive and negative changes between future and historical climates, respectively. Stars mark statistically significant changes ($p < 0.05$).

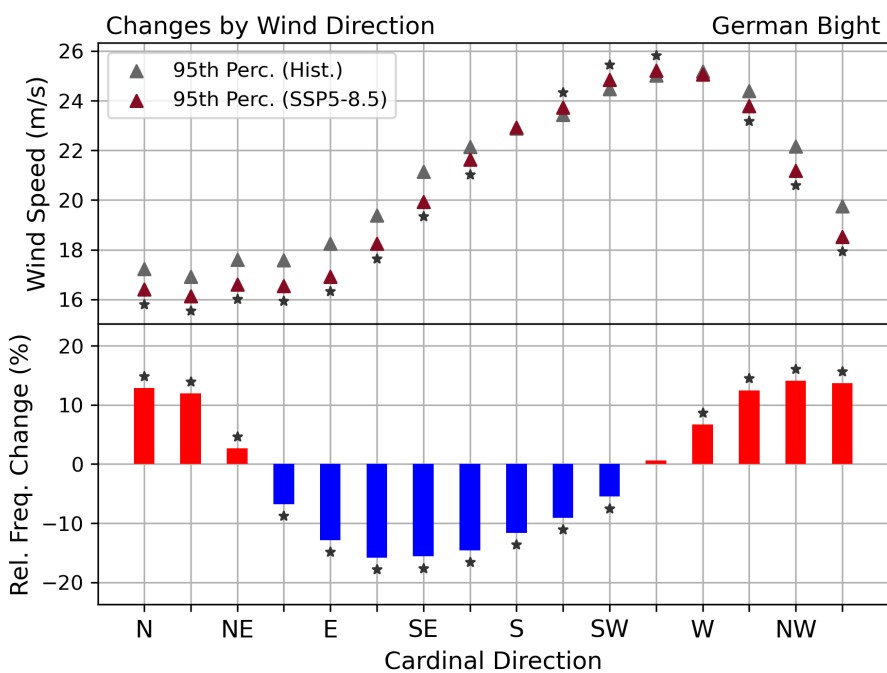

**Figure 7.** (top) Annual 95th percentiles of German Bight geostrophic wind speeds per cardinal direction, averaged over the historical (1961-1990, gray) and the SSP5-8.5 end-of-century climate (2071-2100, maroon). (bottom) Relative frequency changes of annual geostrophic wind directions between the SSP5-8.5 end-of-century (2071-2100) and the historical climate (1961-1990). Data from MPI-GE. Stars mark statistically significant changes ($p < 0.05$).

**Future Risk of Extreme Events**

While the CMIP6 multi-model suite robustly projects decreasing storm activity, i.e., lower 95th percentiles of geostrophic wind speeds, towards the end of the 21st century, both over the German Bight and the Northeast Atlantic, individual extreme events which exceed the 95th percentile can still be a major threat to the population in these areas. The MPI-GE large ensemble with its 50 members for all scenarios allows us to analyze these extreme events in a single-model framework, providing an estimate of the distribution of very high wind speeds in the historical reference climate and showing how the most extreme wind events are likely to change in the projections. Note that there is a larger subset of the CMIP6 models available with three-hourly output than just the MPI-GE. However, we again limit this analysis to the MPI-GE to stay physically consistent within one model, and to avoid inter-model biases that may arise from pooling non-standardized absolute wind speeds from different models.

The distribution of geostrophic wind speeds over the German Bight (Fig. 8) shows that wind speeds between 6 and 10 m/s are the most frequent in both the historical reference period (1961-1990) and the SSP5-8.5 EoC climate (2071-2100), matching the peak in observed wind speeds (1961-1990) as well. While wind speeds below 10 m/s are projected to increase significantly in frequency, wind speeds between 10 and approximately 30 m/s show lower frequencies in the SSP5-8.5 scenario, corresponding to the projected lower storm activity. As a reference, the 95th annual percentiles of geostrophic winds in this region range between approximately 20 and 24 m/s. For very high wind speeds above 40 m/s, however, the EoC climate displays an increase in frequencies, peaking around 50 m/s. Due to the low absolute frequencies of these wind speeds, which correspond approximately to a once in 10-30 years event, changes in frequencies have barely any effect on the 95th percentiles, and are therefore not reflected in the projected storm activity changes. The relative change in frequencies is largest for the most extreme wind speeds (Fig. 9), suggesting that even under lower general storm activity the likelihood for very severe storms may increase. It should be noted that despite the large relative increases in extreme wind speeds, the sample sizes for these events are small and thresholds for statistical significance are higher than for lower wind speeds, as indicated in Fig. 9. A comparison between the geostrophic wind speeds for each percentile (Fig. 10) reveals that despite the increased frequencies of lower wind speeds in SSP5-8.5, the absolute values of lower percentiles are still significantly lower, implying that the overall wind speeds decrease in the EoC climate. Fig. 10 also displays that geostrophic wind speeds above 30 m/s, corresponding to the 99th percentile, are projected to occur more often in the EoC climate than during the historical reference period. The occurrence frequency of 50 m/s events is even expected to triple compared to the historical period.

Similar behavior, i.e., a projected increase in the occurrence frequency of extreme wind events, can be found for some of the Northeast Atlantic triangles as well (Fig. 11). Most of the southern triangles exhibit an increased likelihood for extreme events in the SSP5-8.5 EoC climate, even though some of the triangles show a weakening of lower, less extreme percentiles. The northern triangles, spanning the Norwegian Sea, show an inverse trend, with a reduction in the frequency of very extreme events, accompanied by a reduction of lower percentiles as well.

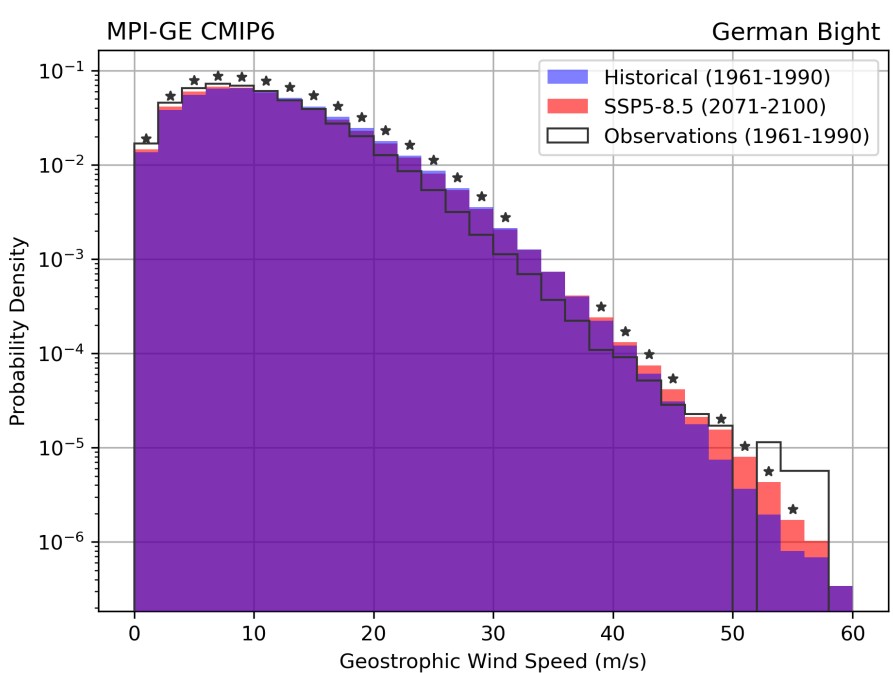

**Figure 8.** Histograms of geostrophic wind speeds in the German Bight in the MPI-GE CMIP6 for the historical period (1961-1990, blue) and the SSP5-8.5 scenario (2071-2100, red), as well as geostrophic wind speeds from observed MSLP measurements (1961-1990, dark gray). Logarithmic y-axis. Stars mark statistically significant changes from historical to SSP5-8.5 ($p < 0.05$).

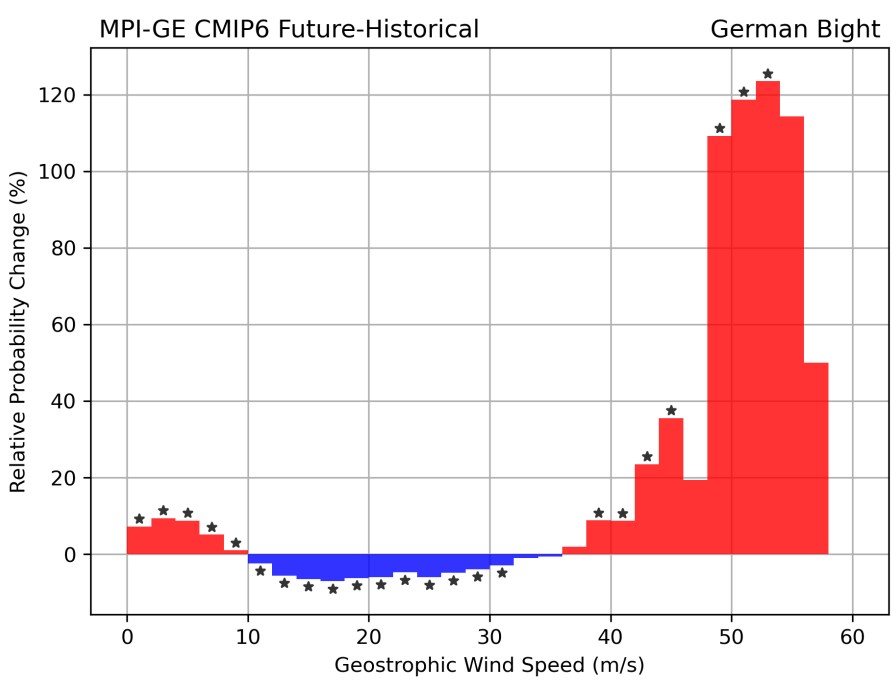

**Figure 9.** Relative probability density differences of geostrophic wind speeds in the German Bight in the MPI-GE CMIP6 between the SSP5-8.5 scenario (2071-2100) and the historical period (1961-1990), i.e. the relative difference between the histograms in Fig. 8. Stars mark statistically significant changes ($p < 0.05$).

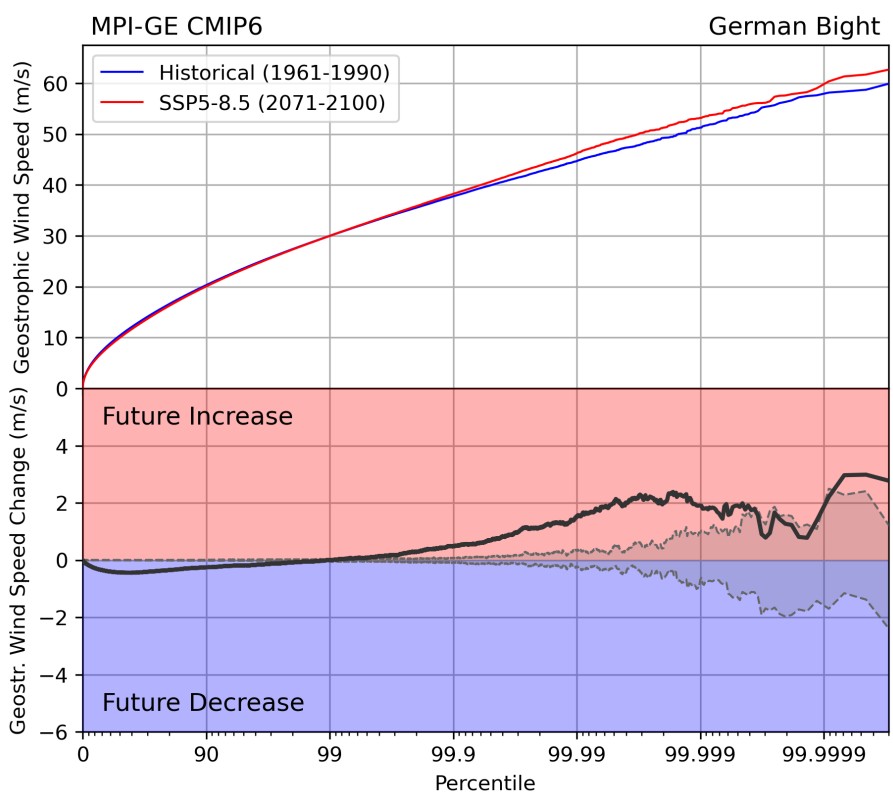

**Figure 10.** Probabilities of geostrophic wind speeds in the German Bight in the MPI-GE CMIP6 for the historical period (1961-1990, blue) and the SSP5-8.5 scenario (2071-2100, red), as well as the difference between SSP5-8.5 and historical (black). Shaded gray areas mark the range of differences that would not be statistically significant ($p < 0.05$). Percentiles refer to the pooled dataset of the entire MPI-GE during the respective time periods, i.e., all timesteps from 30 years and 50 ensemble members. Logarithmic x-axis.

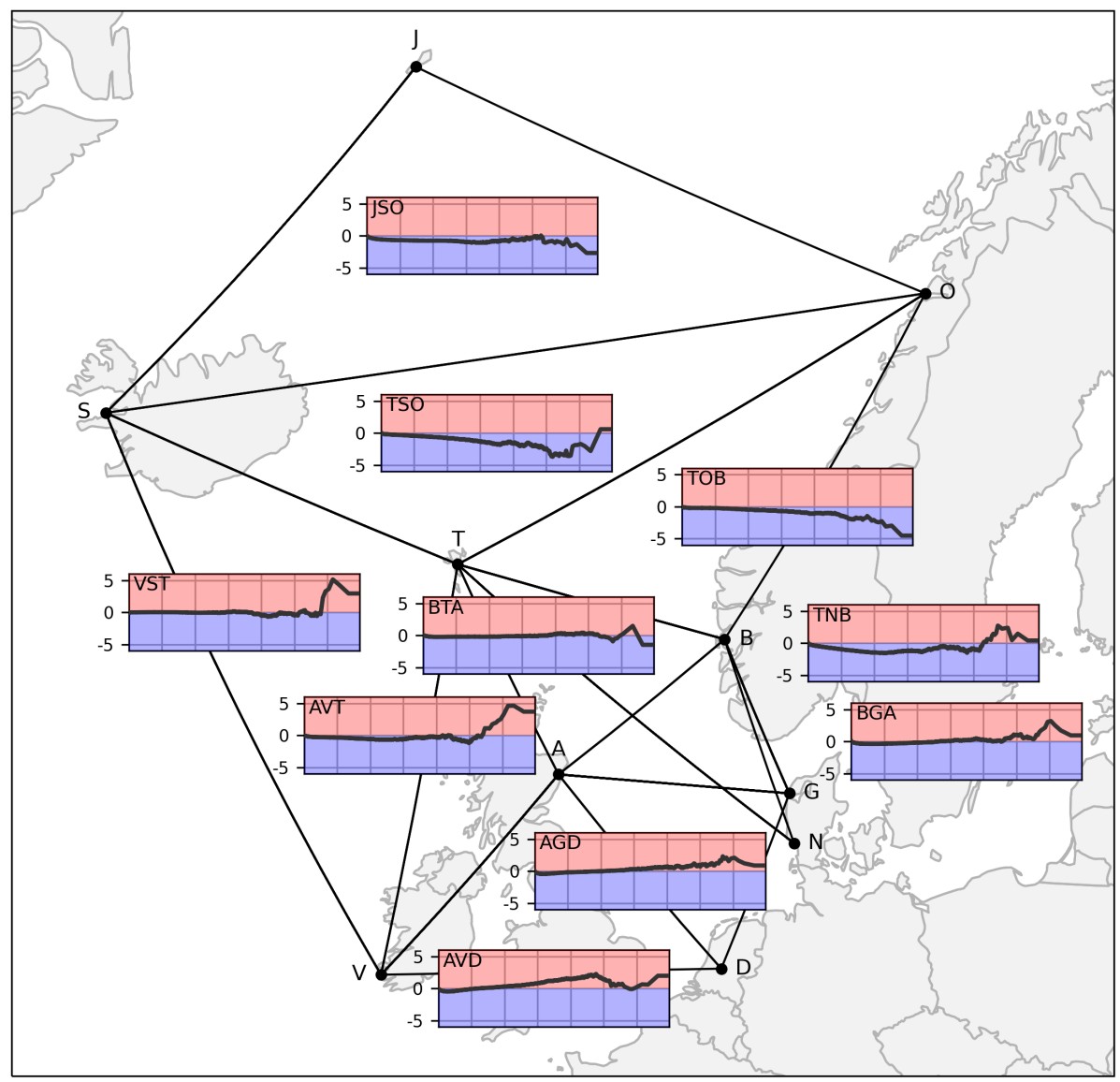

**Figure 11.** Map of the Northeast Atlantic stations and triangles, as well as probability differences of geostrophic wind speeds between SSP5-8.5 and historical for each triangle. Logarithmic x-axis. Axis variables, limits, and data pooling are identical to those in Fig. 10.

## 4 Discussion

We show that storm activity over both the German Bight and the larger Northeast Atlantic Ocean are robustly projected to decrease towards the end of the 21st century by the current generation of global climate models. These findings are somewhat contrary to the results of Harvey et al. (2020), who found a strengthening of the North Atlantic winter storm track over western Europe, based on a multi-model analysis of the winter-mean zonal wind speeds at 250 hPa and the bandpass-filtered variability of mean sea-level pressure (MSLP).

Our analysis uses one commonly used metric for storm activity, the 95th annual percentiles of geostrophic wind speeds derived from horizontal gradients of MSLP. This percentile-based approach combines both the number and intensity of storms integrated over an entire year, but does not explicitly allow for a separate analysis of either the number or the intensity. Therefore, findings like those by Priestley and Catto (2022) who note a decrease in the total number of cyclones, but an increase of very intense cyclones, may not be immediately visible in the percentile-based storm activity index due to the contrasting contributions of the individual factors. Generally, every change in the distribution of wind speeds which does not move the annual 95th percentile will not be detectable by the 95th percentile proxy. In fact, our results for the projected change of the most extreme events for each triangle confirm that future increases or decreases for the uppermost percentiles can be completely independent of those of the 95th or lower percentiles, and that changes in different percentiles also exhibit different spatial patterns. The projected behaviour of the most extreme events, i.e. a reduction in the Norwegian Sea, but an increase over the North Sea and British Isles, is more in line with the storm track changes found by Harvey et al. (2020). Generally, when comparing results of studies on projections of the wind climate, the choice of metric and time period need to be regarded. Even a slight change in, for instance, the integration period (winter season versus calendar year) or the percentile (90th, 95th or 99th) may lead to the metric representing different types of storms and even different drivers and physical mechanisms.

An advantage of the geostrophic proxy is its independence of near-surface wind speeds and their parametrization in the models. While the original motivation behind the use of geostrophic winds was that observational records of MSLP are less inhomogeneous than those of near-surface wind speeds (Schmidt and von Storch, 1993), the MSLP gradient-based proxy also eliminates the error arising from different wind parametrizations among CMIP6 models. Especially when analyzing non-standardized absolute wind speeds, a direct comparison between different models becomes possible with the geostrophic approach. It should be noted however that the geostrophic wind speeds generally overestimate the actual near-surface wind speeds in cyclones.

While our analysis for German Bight storm activity is based on a single triangle, we assess Northeast Atlantic storm activity based on a set of ten mostly non-overlapping triangles, following Alexandersson et al. (1998) and Krueger et al. (2019). We individually compute storm activity for each of the 10 triangles and then average over the entire set. As the storm climate in the respective triangles may be similar but not identical, individual features of certain regions may be smoothed out in the averaging process. The averaging therefore leads to a smaller variability than that of German Bight storm activity, as well as the inability to translate the storm activity values back to absolute geostrophic wind speeds, as the individual 95th percentiles of each triangle are standardized before averaging. Consequently, we have to assess the percentile changes of absolute geostrophic wind speeds

in the final part of our manuscript separately for every triangle. Another consequence of averaging over 10 triangles is the possible loss of distinct features that vary spatially within the Northeast Atlantic region, such as, for example, the weakening of the storm track over the Norwegian Sea, but simultaneous strengthening of the storm track over western Europe as presented by Harvey et al. (2020).

Due to the large range of ensemble sizes between the models participating in CMIP6, our results show sensitivity to the definition and calculation of a multi-model mean. By restricting our bootstrapping to exactly one member from each model regardless of the initial ensemble size, we aim at assigning equal weights to every model. This approach is based on the "one model, one vote" multimodel-mean approach described in Sansom et al. (2013) and Zappa et al. (2013), but uses one randomly selected member per model instead of each model mean. However, we find that this approach underestimates the true uncertainty within the CMIP6 model suite, as approximately half of all models only contribute one member, meaning that half of the bootstrapped ensemble consists of the same fixed time series in every bootstrapped sample. Thus, any estimation of uncertainty can only originate from the remaining half of the models, resulting in an underestimation of the total uncertainty. This discrepancy is especially apparent when single-member models and smaller ensembles, i.e., those with less than 5 members, are discarded (Figs. 2c, d) or when comparing the bootstrapped uncertainty to the standard deviation of the entire set of members (Figs. 2e, f). Also, bootstrapping for multi-member models is done separately for historical experiments and scenarios, but scenario runs may not match their historical counterparts. This can create inconsistencies that obscure climate signals. Ideally, each scenario run would be linked to its historical parent, but data availability prevent this, as the number of available runs varies by model and scenario, and some scenario runs lack a clear historical counterpart. Improved coordination in modeling and data storage could help to resolve these issues. It is therefore imperative to carefully revisit the definitions of multi-model means in comparisons of multi-model studies on the future evolution of storm activity.

The results of this study draw upon the representation of large-scale atmospheric patterns in the Northeast Atlantic on different timescales, which are known to vary strongly between models and are uncertain due to high internal variability (e.g., Deser, 2020). Storm activity in both the Northeast Atlantic and German Bight have been shown to be connected to dominant modes of variability like the North Atlantic Oscillation (NAO) and the Scandinavia Pattern, although this connection appears to be non-stationary (Krueger et al., 2019; Krieger et al., 2021). A recent study by Smith et al. (2025) suggests that deficiencies in the current generation of climate models lead to a systematic underestimation of the true magnitude of variability of the NAO, causing an underestimation of possible extremes in future scenarios, especially in high-emission scenarios. Smith et al. (2025) argue that especially ensemble-mean analyses are affected by these issues. In our results, we also see that the ensemble-mean signal is quite small compared to the internal variability, and that the future evolution of the ensemble mean does not necessarily concur with the projected behavior of severe extreme events.

Our findings for the projected change in wind direction distributions indicate an increase in the likelihood of westerly and northwesterly winds, both in the multi-model and the MPI-GE analyses. Westerly directions are typically associated with certain large-scale circulation types (*Großwetterlagen;* Hess and Brezowsky, 1977) like e.g. Cyclonic West. A recent study be Heinrich et al. (2024) identified a robust climate change signal in the occurrence frequency of Cyclonic West days over Europe in CMIP6 projections, showing a projected increase during winter and decrease during summer. Our results for wind direction

changes confirm the findings of Heinrich et al. (2024), adding that this increase in winter does not necessarily translate to a higher storm activity, as westerly winds are also projected to weaken in intensity.

Building on the findings of this study, promising directions for future research emerge. Systematic seasonal decompositions of storm activity through disaggregation of trends for winter, spring, summer, and fall could uncover shifts in the timing and intensity of storms that are masked by annual averages. This is particularly relevant given the potential for climate change to alter the seasonality of both storm frequency and severity in the North Atlantic region. The application of percentile-based event attribution frameworks could provide quantitative estimates of the changing risk of extreme storm events, connecting large-scale circulation changes to shifts in high-impact wind and pressure events at the regional or local scale. This would also facilitate more robust links between climate model projections and observed weather impacts. Expanding the analysis to explicitly assess compound coastal hazard risks such as the co-occurrence of precipitation and wind-induced storm surges would be highly valuable for impact assessment and adaptation planning, particularly in low-lying coastal areas. Integrating storm activity projections with hydrodynamic and flood models, potentially in conjunction with computationally efficient statistical methods or deep-learning approaches (e.g., Tiggeloven et al., 2021; Schaffer et al., 2025), could clarify the changing likelihood and severity of compound events under future scenarios. Finally, higher-resolution regional climate models or convection-permitting simulations, as they become available for the North Atlantic and adjacent coasts, could help resolve finer-scale storm features, build new reference datasets to downscale our findings to local needs, and foster local adaptation strategies.

## 5   Conclusion

We analyze the evolution of German Bight and Northeast Atlantic storm activity in the CMIP6 multi-model ensemble, as well as the Max Planck Institute Grand Ensemble (MPI-GE), using a well-established proxy based on the 95th annual percentiles of geostrophic winds. In the CMIP6 ensemble, we find a robust downward trend in all scenarios (SSP1-2.6, SSP2-4.5, and SSP5-8.5) for the Northeast Atlantic and a weaker but still downward-facing trend for the German Bight, which we attribute to anthropogenic forcing. Simultaneously, the ensemble projects an increase in westerly and a decrease in easterly winds over the Northeast Atlantic, and an increase in northwesterly and a decrease in southeasterly winds over the German Bight. We show that the MPI-GE generally agrees with the full CMIP6 suite on the projected decline of storm activity, but note a weaker trend in the high-emission SSP5-8.5 scenario, as well as some disagreements between the change in northwesterly wind directions in the German Bight. Using the single-model MPI-GE, we analyze the change in absolute geostrophic wind speeds in the German Bight. We demonstrate that despite an increase in the frequency of westerly and northwesterly winds, the 95th annual percentiles of wind speeds from these directions are projected to decrease, leading to an overall lower storm activity. Moving to even higher percentiles representing the most extreme storm events, however, reveals that the future projections show a strong increase in their frequency in the German Bight and adjacent regions, and a decrease in the northern part of the Northeast Atlantic. We conclude that, while generally we see a downward trend in storm activity-related metrics in future scenarios, especially the most severe storms that currently occur very infrequently, may see a significantly increased likelihood in the future, an evolution that is not captured by many common storm activity metrics.

## Competing interests

The authors declare that they have no conflict of interest.

## Author Contribution

DK and RW conceived and designed the study. DK carried out the analysis and created the figures. DK and RW wrote the manuscript.

## Acknowledgements

DK and RW were supported by the Federal Ministry of Research, Technology and Space (BMFTR) through the projects "METAscales" (FKZ 03F0955J) and "WAKOS – Wasser an den Küsten Ostfrieslands" (FKZ 01LR2003A), as well as the northern German states within the scope of the German Marine Research Alliance (DAM) mission "mareXtreme". DK received funding from BMFTR through the project "A Coming Decade - Decadal climate predictions for Europe" (FKZ 01LP2327A) within the framework of the Strategy "Research for Sustainability" (FONA) RW received funding by the BMFTR through the project "ECAS-Baltic" (FKZ 03F0860C).

We acknowledge the World Climate Research Programme (WCRP), which, through its Working Group on Coupled Modelling, coordinated and promoted CMIP6. We thank the climate modeling groups for producing and making available their model output, the Earth System Grid Federation (ESGF) for archiving the data and providing access, and the multiple funding agencies who support CMIP6 and ESGF.

We also thank the German Climate Computing Center (*Deutsches Klimarechenzentrum*; DKRZ) for enabling this study by providing computational resources.

## Data availability

The simulations of MPI-GE CMIP6 can be accessed via DKRZ's ESGF server at https://esgf-data.dkrz.de/search/cmip6-dkrz/ by specifying "Source ID: MPI-ESM1-2-LR", "Institution: MPI-M' "Experiment: historical/ssp126/ssp245/ssp585", "Frequency: 3hr" and "Variant Label: rXi1p1f1" with X ranging from 1 to 50.

Observed timeseries of Northeast Atlantic and German Bight storm activity based on Krueger et al. (2019) and Krieger et al. (2021) can be found under Krieger (2025).

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
