# Peer review of "CMIP6 Multi-model Assessment of Northeast Atlantic and German Bight Storm Activity"

_EGUsphere, 2025_

## Author Comment (AC1)

**Response to Reviewer Comment RC1**

We sincerely thank Reviewer #1 for their constructive and insightful comments on our manuscript *CMIP6 Multi-model Assessment of Northeast Atlantic and German Bight Storm Activity*. The comments greatly helped us to improve the manuscript and clarify key points.

We respectfully acknowledge the reviewer's concern regarding the scope of the journal. However, we would like to point out that the editor, having considered the submission in light of the journal's aims and scope, deemed it suitable for peer review. We believe this reflects the editor's judgment that the manuscript aligns with the journal's thematic focus, which has recently also included studies of regional significance (e.g., https://esd.copernicus.org/articles/special_issue1088.html). In this context, we trust that decisions regarding scope remain within the editorial purview, while the peer review process can focus on the scientific quality, clarity, and contribution of the work.

In the following, we will give a point-by-point response to the reviewer's comments and describe how we plan to address the issues raised.

**Main comments:**

**1** My main concern is that as presented, the study appears rather incremental. There are many studies examining model projections of North Atlantic storminess (as you summarise in your introduction), and your key conclusion of an overall reduction in storminess in future model projections but with an increase in the intensity of the most extreme storms, has been noted numerous times before. Please tweak the framing of your work to address this concern (particularly in the introduction) to better inform the reader exactly how this study aims to advance current understanding. Formulating one or two explicit research questions might help with this.

**Response:** We thank the reviewer for this comment. We agree that the framing of the manuscript can be improved to better convey the novelty of our approach. We will revise the introduction by formulating explicit research questions and emphasize our twofold novelty: (1) the combined use of CMIP6 multi-model-ensemble output and the 50-member MPI-GE to disentangle externally forced signals from internal variability, and (2) the inclusion of the pressure-based storm activity proxy. The number of CMIP6 models used in previous studies on storm activity changes has been limited, primarily due to the unavailability of key diagnostic variables across all models. To overcome this constraint, we apply the pressure-based proxy introduced by Schmidt and von Storch (1993), which enables the inclusion of a larger ensemble of 32 CMIP6 models. This broader model set allows for a more comprehensive assessment of projected changes and uncertainties in Northeast Atlantic storm activity under various anthropogenic forcing scenarios, as well as a direct comparison with observed pressure-based storm activity. These additions will clarify the contribution of our work to the ongoing debate on storm activity projections and their uncertainties.

**2** To my mind, one key advance is the comparison of storminess between the climate models and the long-term dataset of direct observations, because the vast majority of climate model studies just compare against reanalysis products. However, this comparison is not mentioned in the abstract, and even a basic description of the observational dataset is omitted from the manuscript. I'd urge you to make more of this aspect in the text, and to extend the observational comparison to all relevant figures (e.g. 3, 4, 6, 7, 8) if possible.

**Response:** We thank the reviewer for highlighting this important aspect. We agree that this comparison deserves more emphasis. We will revise the abstract to explicitly mention the observational dataset. In the main text, we will introduce and describe the observational data in Section 2 and extend the comparison to all relevant figures, including Figs. 3, 4, 6, 7, and 8, where possible.

**3** Your storminess diagnostics are annual in the sense that you don't subset the data to a particular season. However, I imagine most of the >95%ile geostrophic wind events happen in autumn/winter and so your projected future changes represent most closely the changes in these seasons. Given projected future

changes in storminess contain important seasonal variations, please add a discussion on this point to aid interpretation.

**Response:** We thank the reviewer for raising this point. We will add a discussion paragraph to Section 5 that clarifies the seasonality of the geostrophic storm activity metrics. We will state that while we use annual metrics for comparability and simplicity, the most extreme geostrophic wind events are indeed most frequent during late autumn, winter, and early spring. Earlier studies have shown that the geostrophic wind statistics of those seasons indeed closely resemble the annual statistic (e.g. Krieger et al., 2021). We will also add that the lack of seasonal disaggregation may mask more nuanced seasonal shifts in storm activity under climate change. Also, many studies do not focus on seasonal shifts or aspects, so that analyzing annual values improves comparability with other literature.

**Other comments:**

**Abstract** The last two sentences appear contradictory because you state "the upper percentiles of winds speeds from these directions decrease" and then "the most extreme storms may become stronger or more likely". I think the former is referring to the 95th percentile of the wind speeds whereas the latter is referring to more extreme percentiles. Please clarify.

**Response:** We thank the reviewer for bringing this inconsistency to our attention. Indeed, the former is referring to the 95$^{th}$ percentile of the wind speeds per wind direction, and the latter to the more extreme percentiles. We will rephrase the abstract to remove this confusing wording.

**L25 and the following paragraphs** Please be explicit about the seasonality of the projected changes in storminess presented in these papers. Some I know explicitly refer to winter only, and others I am not sure about.

**Response:** We will revise the introduction to specify which studies refer to seasonal/winter-only projections and which focus on annual statistics to provide better context for the reader.

**L56** "upper wind speed percentiles" is unclear (I thought it meant upper-tropospheric wind speeds initially). Please clarify, e.g. "upper percentiles of near-surface wind speeds". Similar comment applies to L58.

**Response:** We apologize for the misleading wording and see the issue with the term "upper wind speed percentiles". We will adapt the wording as suggested.

**L91** Is CMIP6 psl data daily means or instantaneous?

**Response:** The daily MSLP data consists of daily means, not instantaneous values. We will add this information to the data description.

**L97** Just to be clear, do you standardise the annual 95th percentiles for each triangle separately, or average them together and then standardise?

**Response:** We standardize each triangle separately, and then average over the standardized timeseries of all triangles. We repeat this step for every member in the ensemble. We will add a clarifying statement to the respective section.

**L115** I presume that the gradients are computed using the distances between the model grid points (which differ for each model), rather than the original station locations? Please specify.

**Response:** That is correct, all gradients are computed based on the locations of the respective model gridpoints. We will clarify this part to avoid confusion.

**L133** The observed timeseries has not been introduced. Please add a description of it in section 2.

**Response:** We will add a description of the storm activity observations in the Methods and Data section, specifying the data sources, spatial coverage, time range, and method used for storm activity estimation.

**L146** You claim that "the full pool of ensemble members can represent the variability present in the observations", but this is misleading and clearly must depend on the timescale examined. If I understand correctly, all the timeseries are independently standardised, so the interannual variability is by construction captured by the ensemble, at least during the period 1960-1990. What you show are ten year running means, so I assume your claim is something like "the full pool of ensemble members can represent the variability on decadal timescales". Please clarify.

**Response:** The reviewer is correct in their assumption. By showing the standard deviation of the full pooled ensemble and the observed storm activity as 10 year running means, we demonstrate that the decadal variability of observed storm activity is contained within the multi-model ensemble. While this is correct by construction for the reference period of 1961-1990, it also holds for periods after and before the reference period. We will rewrite this section to avoid further misconceptions and make clear that this figure does not show the interannual variability, but rather the variability on decadal and longer timescales.

**Section 4** Several recent papers have highlighted deficiencies in the ability of climate models to simulate multi-decadal variability in the North Atlantic, and have questioned the reliability of model projections in this region as a result (e.g. see here, and references therein: Smith et al., 2025, https://doi.org/10.1038/s41558-025-02277-2). Given their importance, I'd urge you to extend your discussion to include reference to them and relate to the findings of your study.

**Response:** We thank the reviewer for bringing up the recent publication by Smith et al. The study is highly relevant to our discussion and we will thus expand our discussion section to reflect more on the points raised by Smith et al. (2025) and related papers.

**L225** "increase" -> "increase in frequency"

**Response:** We will update our wording here.

**L290** This paper presents a statistical methodology for assessing future changes from multi-model ensembles of differing sizes, which is very relevant to your suggestion: Zappa et al. (2013) A multimodel assessment of future projections of North Atlantic and European extratropical cyclones in the CMIP5 climate models. Journal of Climate, 26(16), pp.5846-5862.

**Response:** We appreciate the reviewer for pointing us to the study by Zappa et al. (2013). While we refer to this study in the introduction, we absolutely see the need to mention it in this paragraph as well. We will update this part of the discussion to include this study.

**Fig 3 caption** Please state the periods over which trends are computed (I assume the full experiment periods, but best to be precise).

**Response:** Yes, the trends are computed across the entire experiment length. We will add that to the figure caption.

**Fig 4 caption** Are daily geostrophic wind directions? Please clarify.

**Response:** Yes, the wind directions in Fig. 4 are daily means, as they are based on daily-mean MSLP input data. The wind directions in Fig. 6 are three-hourly. We will add the frequencies to the figure captions.

**Fig 7 caption** Repeated "the"

**Response:** We thank the reviewer for spotting this mistake. We will correct it in the revised version.

**Fig 8** To what extent are the differences here statistically robust? Can you construct confidence intervals? (here and/or Fig 9)

**Response:** We thank the reviewer for bringing up the issue of missing statistical significance checks. In accordance with comments by Reviewer #2, we will add estimates of robustness to the manuscript and figures wherever necessary.

We, the authors, would like to thank Reviewer #1 again for their careful reading of our manuscript and for the constructive comments. We hope that our responses and proposed revisions clarified all outstanding points and look forward to further feedback.

With kind regards,

Daniel Krieger and Ralf Weisse

---

## Author Comment (AC2)

**Response to Reviewer Comment RC2**

We sincerely thank Reviewer #2 for their constructive and insightful comments on our manuscript *CMIP6 Multi-model Assessment of Northeast Atlantic and German Bight Storm Activity*. The comments greatly helped us to improve the manuscript and clarify key points.
In the following, we will give a point-by-point response to the reviewer's comments and describe how we plan to address the issues raised.

**General comments:**

**A** A number of analyses in this study lack any assessment of robustness, e.g. by checking for statistical significance or provision of confidence intervals. In particular this is the case for results presented in Figs. 4, 6, 7, 8, 9, 10, and 11. I would be willing to accept this in case of Fig. 11 in order to ensure readability but for the other Figs. an analysis of statistical significance (nothing mentioned in the text either regarding these results) as well as including any indication in this respect in the Figs. seems necessary and possible. The authors themselves discuss the strong sensitivity of such studies depending on choice of metric, integration period, storm identification method, ... (lines 259-262). Especially in these cases, a thorough assessment of robustness and statistical significance is unavoidable for a proper scientific study.

**Response:** We agree with the reviewer that the mentioned figures require an assessment of significance or confidence. We will reproduce the figures from the analysis with added indicators of statistical significance. We will also expand the results and discussion section with thorough analysis of the significance test and implications for our conclusions.

**B** The authors introduce very briefly, why the analysed parameters matter. Apart from lines 14-19 which sketch a few possible impacts of Northeast Atlantic storms, there is nothing. In particular, they provide no reasoning why wind direction is interesting besides wind speed/storm activity alone. A number of reasons are very clear to me but readers not familiar with the specific of potential storm impacts will have no clue why wind direction is relevant beyond academic exercise.

**Response:** We thank the reviewer for this observation. We will revise the introduction to include a paragraph that motivates the relevance of wind direction analysis. In particular, we will emphasize its importance for understanding the pathways of storm systems, storm surge impacts on coastal areas, directional wind stress on infrastructure, and compound flooding risks due to onshore winds. Furthermore, wave height and direction are dependent on the wind direction (via the wind fetch), and play a big role in determining coastal impacts such as erosion. This will help clarify the added value of this analysis for readers unfamiliar with storm impact mechanisms.

**C** I am a little worried regarding the compilation of the multi-model ensemble. The authors themselves stress a problem related to their procedure of selecting one ensemble member from each model which leads to thos models with just one member always remaining part of the multi-model ensemble. This is a major shortcoming of this study. I am grateful that the authors have the courage to outline this shortcoming themselves. This is a great example of good scientific practice and they present an approach to asses the potential influence by compiling a second multi-model ensemble, bootstrapping only from thos models with 5 or more members. Anyway, I am worried for one more reason: If I understand correctly, the authors perform the bootstrapping for those models with more than one member separately for the historical experiment and the individual scenarios. When doing so, a chosen scenario run is probably unrelated to its historical counterpart (at least in most cases). This yields inhomogeneities masking physical meaningful climate signals. In the end, it probably does not matter so much, given that the averaging over the multii-model ensemble is done, smoothing out this inconsistency between the end of the historical period and three separate beginnings of future scenarios. However, I would argue that choosing scenario simulations belonging to the chosen historical parent simulation had been a better solution. If my understanding is correct, I would ask the authors to include this issue in the discussion, too.

**Response:** We greatly appreciate this comment. The reviewer is correct in assuming that the bootstrapping is performed separately for the historical and the scenario runs. The main reason why we did not choose the same runs for historical and scenarios is that not all members continue from the historical to the

scenario period. In other words, the number of available runs changes between historical and the different scenarios for certain models, and some scenario runs do not have a clear parent in the historical period. A way to circumvent this is to only allow for the selection of those members where a scenario run is clearly connected to a historical parent run. We will investigate the feasibility of this approach and add a discussion of this issue in Section 4, including its potential implications for model consistency and the physical realism of trend transitions.

**D** lines 113-115: Choosing the nearest grid point from each model for a given observational site may lead to some distortion in areas of steep orographic gradients, namely Bodoe and Bergen. I understand that the authors use SLP, however, an extrapolation of modelled surface pressure into the ground of the one model (where the closest grid point is inland) may be somewhat different from SLP practically identical to modelled surface pressure of another model (where the closest grid point is flat terrain or even ocean). I do not insist a priori on including a discussion of this matter in the manuscript but I challenge the authors to think about this possible case (or perform some sensitivity test) and provide arguments here, why this should not be relevant for their findings.

**Response:** We thank the reviewer for this thoughtful comment. We will add a brief discussion in the methods section acknowledging that the selection of the nearest model grid point may introduce slight distortions in orographically complex regions such as Bodø and Bergen. We will mention this explicitly in the revised manuscript.

**E** I am pretty confused by the section title of Sec. 3.2. "Internal variability" refers to temporal fluctuations of various variables inherent to the natural earth system, observed or modelled. Single-model initial condition large ensembles (SMILE) are great tools to distinguish extrenally forced signals from internal variability. However, this is not what you are analysing in the large part of this section. Instead, the main focus of this section are heterogeneous climate change signals for different parts of the storm intensity spectrum as well as depending on wind direction. I would suggest choosing a more suitable title for Sec. 3.2 plus deleting a few sentences in the first paragraph of Sec. 3.2 (see my specific comment below).

**Response:** We appreciate this concern regarding the title of Section 3.2. In accordance to our response to the specific comment below, we will completely rework the first part of this section so that it does not mix up changes in the ensemble mean and internal variability anymore. We will also follow the reviewer's suggestion and rename the section to better reflect its focus on future changes in the extreme tails of the wind speed distribution.

**Specific comments:**

**L26-28** Actually, the synthesis of Feser et al. is the other way around (increasing north of 60°N, decreasing south of 60°N) in line with a poleward shift of the NH storm track.

**Response:** We thank the reviewer for catching this error. Indeed, the synthesis of Feser et al. concludes that a majority of studies project decreasing storm activity south of 55-60°N and increasing storm activity north of that. We will correct this error in the revised version.

**L38-39** I don't understand the line of argumentation here. The previous sentence is about low model agreement, hence large model-related uncertainty. This sentence here now seems to present a link to substantial changes in wind extremes when combined with changes in storm track locations. It seems something is missing here in between.

**Response:** We intended to argue that the large model-related uncertainty in both the location/shift of the storm tracks and the cyclone density leads to a very high uncertainty in the future evolution of (extreme) wind speed distributions at certain locations in the North Atlantic sector. We will rephrase this paragraph to clarify our line of argumentation.

**L97** Did the authors analyse if discrepancies are introduced when using 3-hourly data for MPI-ESM compared to daily averages for the other models? An average of the eight 3-hourly values is usually in quite

good agreement with a respective daily mean. So, it might even haven been an alternative to calculate such a proxy daily mean for MPI-ESM before analysing its simulations consistently with the other models.

**Response:** We appreciate this concern. For the multi-model analysis, we used daily-mean pressure data only, also from the MPI-ESM model. The three-hourly data are used exclusively in the single-model large-ensemble analysis. We see that this is not explained clearly enough in the Methods section and will revise the respective paragraphs accordingly. We agree that comparing storm activity values calculated from three-hourly data with those calculated from daily values would introduce inconsistencies.

**Sec. 2.2** It is not entirely clear from this section if MPI-GE is also used as part of the CMIP6 multi-model ensemble and hence included in respective analyses or not.

**Response:** We apologize for the unclarity in this regard. The MPI-GE is part of the CMIP6 multi-model ensemble. In the multi-model sections, we use daily-mean SLP data from all listed CMIP6 models, including the MPI-GE (MPI-ESM-LR). We see that assigning two names to the same model (MPI-GE and MPI-ESM-LR) might be confusing to the reader, and will improve the explanation and use of the terms MPI-GE and MPI-ESM-LR accordingly.

**Sec. 2.3** This section lacks the information that an averaging over all ten triangles is performed to yield results for the Northeast Atlantic. This fact is explicitly written only in lines 271-272, that is the discussion.

**Response:** We agree with the reviewer that this important step is missing here. We will add a paragraph on the averaging that is performed over the ten triangles.

**L115-117** Sory, I don't understand the procedure for "triangles" that basically fall onto a line. I am lost when you write about "the observation site that is most distant to the corresponding gridpoint". Which gridpoint? It is three observation sites with three associated grid points... Please rephrase (or correct?) your description here.

**Response:** We apologize for the lack of clarity in this section and thank the reviewer for pointing it out. The procedure aims to construct a triangular area for each model that mirrors the geometry of the three observational sites used to estimate storm activity. For each model, we identify the grid points that are geographically closest to the three observational stations. These three grid points define a triangle over which the geostrophic wind is calculated.

However, in some models, the three closest grid points can fall nearly on a straight line (e.g., sharing the same latitude or longitude), which would prevent a meaningful calculation of the wind vector due to an enclosed area of zero. In such cases, we slightly adjust the position of one grid point to form a proper triangle. Specifically, we move the grid point corresponding to the observation site that is geometrically furthest from the initially assigned grid point. This adjustment is limited to a single grid cell in the nearest orthogonal direction to preserve the original geometry as closely as possible while ensuring a valid triangle. We will revise the manuscript to explain this procedure more clearly.

**L125&149**: I would refrain from using the term "multidecadal oscillation" here. For me this term implies some type of natural variability associated with these fluctuations. But this is not possible given that you analyse the ensemble mean (and certain ensemble quantiles) here. These fluctuations must be either by chance (unlikely in these cases) or also result from external forcing common to all individual simulations. If you discuss these fluctuations, you have to provide possible drivers here.

**Response:** We agree that the term "multidecadal oscillation" is inappropriate here as the ensemble mean can only reflect variability that relates to an external forced climate signal. We will rewrite the respective sentences to avoid suggesting that the ensemble mean variability hints at an oscillation originating from within the Earth system.

**Fig. 3** I would suggest swapping subfigures vertically in order to present the same order as in previous figures: first, the results from the bootstrapped MME, second, the results from all members.

**Response:** We will follow the reviewer's suggestion and rearrange the subfigures.

**L178-179** "Internal variability" refers to fluctuations inherent to the natural earth system, observed or modelled. Single-model initial condition large ensembles (SMILE) are great tools to distinguish extrenally forced signals from internal variability. However, you cannot diagnose the internal variability from the SMILE's ensemble mean. It seems to me that you are doing that here. I would suggest eliminating everything from line 178 to line 196, replacing that with a better introduction for the direction- and intensity-dependent analyses, and then continue with line 197.

**Response:** We thank the reviewer for bringing this up. We agree that the ensemble mean of the SMILE is not a suitable tool to quantify internal variability. We see the need to disentangle this paragraph into a part on the general evolution of the SMILE ensemble mean across the scenarios (i.e., the forced signal), and the actual analysis on the internal variability (extremes, wind directions) which uses the pooled ensemble data and not just the ensemble mean. We will take up the reviewer's suggestion to completely rewrite the first part of this section.

**L201** It looks to me as if the (positive) frequency changes are rotated clockwise (not counter-clockwise) compared to the historical frequencies.

**Response:** The reviewer is correct that the positive (red) frequency changes are rotated clockwise compared to the historical (gray) frequencies. However, in this sentence we refer to the differences in frequency changes between the MPI-GE (Figure 6b) and the CMIP6-MME (Figure 4b), which show a counterclockwise shift from CMIP6-MME to MPI-GE. We will rephrase this part of the results to clarify which wind roses we compare to avoid misunderstanding.

**L206-208 ff.** Why are the changes of upper percentiles of wind speeds only analysed from MPI-GE. This could have been done from the CMIP6-MME as well or not?

**Response:** The reason we performed the extreme wind percentile analysis only for MPI-GE is the availability of high temporal resolution data (3-hourly), combined with the large ensemble size within a single model, i.e. with consistent model physics. Most CMIP6 models only provide daily data, which is less sufficient for robust storm activity estimation at the high end of the distribution. We will clarify this constraint in the manuscript.

**L210** Here you write about "significantly weaker". Have you checked statistical significance? If not, refrain from using this term here but as said in my general comment A) you will have to assess statistical significance.

**Response:** We agree that the term "significantly" should not be used without supporting statistical tests. In line with our response to general comment A, we will include significance tests for these results and update the language and figures appropriately.

**L219-222** Why does only MPI-GE allow for the analysis of the extreme events? You could have done the same with the full CMIP6-MME, or not?

**Response:** Similar to the point above, we focus on MPI-GE for extreme event analysis due to its temporal resolution and ensemble size. However, we acknowledge that a subset of CMIP6 models could be used for similar analysis if suitable data are available. We will mention this limitation explicitly.

**L304-305** Rephrase this sentence about the internal variability. This is not what you are looking at.

**Response:** We will remove the term "internal variability" here and rewrite the sentence to reflect that the ensemble spread is able to encompass the observed variability.

**Outlook** I would suggest at least one or two outlook sentences... What are possible next steps that could be done based on your findings?

**Response:** We thank the reviewer for this suggestion. We will add a short outlook section at the end of the discussion, highlighting potential next steps such as including a seasonal decomposition, applying percentile-based event attribution approaches, and investigating compound coastal hazard risks under future storm conditions.

**Technical corrections:**

**L52** Insert "that is" before "most pronounced in the CMIP3...".

**Response:** We will reword this sentence as suggested by the reviewer.

**L110** The reference to Krueger et al. (2019) makes no sense here. It is correctly placed in line 111, so please delete it here.

**Response:** We agree and will delete the reference in line 110.

**L277** Replace "second part" by "final part". It's just this last element and not half of your study.

**Response:** We will replace the term as suggested.

We, the authors, would like to thank Reviewer #2 again for their careful reading of our manuscript and for the constructive comments. We hope that our responses and proposed revisions clarified all outstanding points and look forward to further feedback.

With kind regards,

Daniel Krieger and Ralf Weisse

---

## Author Response (AR1)

Response Letter to 1st Round of Reviews - egusphere-2025-111

We sincerely thank the two anonymous reviewers for their constructive and insightful comments on our manuscript *CMIP6 Multi-model Assessment of Northeast Atlantic and German Bight Storm Activity*. The comments greatly helped us to improve the manuscript and clarify key points.

We greatly appreciate the handling editor Jadranka Sepic for giving us the opportunity to submit a revised manuscript, incorporating the suggestions and comments raised by the two reviewers.

In the following, we will give a point-by-point response to the reviewers' comments and describe how we addressed the issues raised in the revised version.

**Response to RC1**

**Main comments:**

**1** My main concern is that as presented, the study appears rather incremental. There are many studies examining model projections of North Atlantic storminess (as you summarise in your introduction), and your key conclusion of an overall reduction in storminess in future model projections but with an increase in the intensity of the most extreme storms, has been noted numerous times before. Please tweak the framing of your work to address this concern (particularly in the introduction) to better inform the reader exactly how this study aims to advance current understanding. Formulating one or two explicit research questions might help with this.

**Response:** We thank the reviewer for this comment. We agree that the framing of the manuscript can be improved to better convey the novelty of our approach. We revised the introduction to emphasize the two research questions and our two novelties: the use of the geostrophic proxy and full CMIP6 ensemble size, as well as the high-frequency SMILE of the MPI-GE that allows us to examine the most extreme events and shifts in the wind speed distributions. We also removed the mentioning of separating internal variability from the forced response in MPI-GE as we are not focusing on that in the study.

**2** To my mind, one key advance is the comparison of storminess between the climate models and the long-term dataset of direct observations, because the vast majority of climate model studies just compare against reanalysis products. However, this comparison is not mentioned in the abstract, and even a basic description of the observational dataset is omitted from the manuscript. I'd urge you to make more of this aspect in the text, and to extend the observational comparison to all relevant figures (e.g. 3, 4, 6, 7, 8) if possible.

**Response:** We thank the reviewer for highlighting this important aspect. The abstract was revised to explicitly mention the observational dataset. In both the abstract and the main text, the observational data are introduced and described, and a comparison has been added to the Figures where applicable.

**3** Your storminess diagnostics are annual in the sense that you don't subset the data to a particular season. However, I imagine most of the >95%ile geostrophic wind events happen in autumn/winter and so your projected future changes represent most closely the changes in these seasons. Given projected future changes in storminess contain important seasonal variations, please add a discussion on this point to aid interpretation.

**Response:** We thank the reviewer for raising this point. We expanded the discussion to clarify the impact of seasonality of the geostrophic storm activity metrics. We discuss the sensitivity of storm activity to the choice of season and percentile (among others), explain how winter and annual storm activity bear close resemblance, and why we chose to stick the annual view to uphold comparability to the majority of existing literature.

**Other comments:**

**Abstract** The last two sentences appear contradictory because you state "the upper percentiles of winds speeds from these directions decrease" and then "the most extreme storms may become stronger or more likely". I think the former is referring to the 95th percentile of the wind speeds whereas the latter is referring to more extreme percentiles. Please clarify.

**Response:** We thank the reviewer for bringing this inconsistency to our attention. Indeed, the former is referring to the 95$^{th}$ percentile of the wind speeds per wind direction, and the latter to the more extreme percentiles. We rephrased the abstract to incorporate these suggestions.

**L25 and the following paragraphs** Please be explicit about the seasonality of the projected changes in storminess presented in these papers. Some I know explicitly refer to winter only, and others I am not sure about.

**Response:** We will revise the introduction to specify which studies refer to seasonal/winter-only projections and which focus on annual statistics to provide better context for the reader.

**L56** "upper wind speed percentiles" is unclear (I thought it meant upper-tropospheric wind speeds initially). Please clarify, e.g. "upper percentiles of near-surface wind speeds". Similar comment applies to L58.

**Response:** We changed "upper wind speed percentiles" to "upper percentiles of near-surface wind speeds" and "upper geostrophic wind speed percentiles" to "upper percentiles of geostrophic wind speeds"

**L91** Is CMIP6 psl data daily means or instantaneous?

**Response:** Both three-hourly and daily MSLP data consist of temporal means, not instantaneous values. We added this information to the data description.

**L97** Just to be clear, do you standardise the annual 95th percentiles for each triangle separately, or average them together and then standardise?

**Response:** We standardize each triangle separately, and then average over the standardized timeseries of all triangles. We repeat this step for every member in the ensemble. We added a clarifying statement to the respective section.

**L115** I presume that the gradients are computed using the distances between the model grid points (which differ for each model), rather than the original station locations? Please specify.

**Response:** That is correct, all gradients are computed based on the locations of the respective model gridpoints. We added a note on this to the methods section.

**L133** The observed timeseries has not been introduced. Please add a description of it in section 2.

**Response:** We added descriptions of the observational time series of storm activity to the Methods and Data section.

**L146** You claim that "the full pool of ensemble members can represent the variability present in the observations", but this is misleading and clearly must depend on the timescale examined. If I understand correctly, all the timeseries are independently standardised, so the interannual variability is by construction captured by the ensemble, at least during the period 1960-1990. What you show are ten year running means, so I assume your claim is something like "the full pool of ensemble members can represent the variability on decadal timescales". Please clarify.

**Response:** The reviewer is correct in their assumption. By showing the standard deviation of the full pooled ensemble and the observed storm activity as 10 year running means, we demonstrate that the decadal variability of observed storm activity is contained within the multi-model ensemble. We changed this

paragraph to explicitly state that the decadal-scale variability is captured by the pooled ensemble and that this is not only true for the reference period 1961-1990, but also the time before and after.

**Section 4** Several recent papers have highlighted deficiencies in the ability of climate models to simulate multi-decadal variability in the North Atlantic, and have questioned the reliability of model projections in this region as a result (e.g. see here, and references therein: Smith et al., 2025, https://doi.org/10.1038/s41558-025-02277-2). Given their importance, I'd urge you to extend your discussion to include reference to them and relate to the findings of your study.

**Response:** We thank the reviewer for bringing up the recent publication by Smith et al. The study is highly relevant to our discussion and we added a paragraph to reflect on the points raised by Smith et al. (2025) regarding the underestimation of true variability in the North Atlantic.

**L225** "increase" -> "increase in frequency"

**Response:** We updated the wording.

**L290** This paper presents a statistical methodology for assessing future changes from multi-model ensembles of differing sizes, which is very relevant to your suggestion: Zappa et al. (2013) A multimodel assessment of future projections of North Atlantic and European extratropical cyclones in the CMIP5 climate models. Journal of Climate, 26(16), pp.5846-5862.

**Response:** Thank you for highlighting Zappa et al. (2013). Although we cite this study in the introduction, we recognize its relevance here and have revised the paragraph to include their approach in our discussion.

**Fig 3 caption** Please state the periods over which trends are computed (I assume the full experiment periods, but best to be precise).

**Response:** Yes, the trends are computed across the entire experiment length. We added this information to the figure caption.

**Fig 4 caption** Are daily geostrophic wind directions? Please clarify.

**Response:** Yes, the wind directions in Fig. 4 are daily means, as they are based on daily-mean MSLP input data. The wind directions in Fig. 6 are three-hourly. We added the frequencies to both figure captions.

**Fig 7 caption** Repeated "the"

**Response:** We corrected this mistake.

**Fig 8** To what extent are the differences here statistically robust? Can you construct confidence intervals? (here and/or Fig 9)

**Response:** We thank the reviewer for bringing up the issue of missing statistical significance checks. In accordance with comments by Reviewer #2, we added estimates of statistical significance to the manuscript and figures. In this case, the changes in Figures 8 and 9 are significant at the 0.05-level for most bins except the two most intense (>54 m/s) and those close to the wind speed where the sign changes from negative to positive (32-38 m/s).

Response to RC2

**General comments:**

**A** A number of analyses in this study lack any assessment of robustness, e.g. by checking for statistical significance or provision of confidence intervals. In particular this is the case for results presented in Figs. 4, 6, 7, 8, 9, 10, and 11. I would be willing to accept this in case of Fig. 11 in order to ensure readability but for the other Figs. an analysis of statistical significance (nothing mentioned in the text either regarding these results) as well as including any indication in this respect in the Figs. seems necessary and possible. The authors themselves discuss the strong sensitivity of such studies depending on choice of metric, integration period, storm identification method, ... (lines 259-262). Especially in these cases, a thorough assessment of robustness and statistical significance is unavoidable for a proper scientific study.

**Response:** We agree with the reviewer that the mentioned figures require an assessment of significance or confidence. We added estimates of statistical significance to the aforementioned figures and discuss them in the manuscript. The assessment of significance uses a classical bootstrapping approach to estimate the possible range of outcomes through repeated sampling. We updated figures and text in the revised manuscript, and added a description of the bootstrapping to the Methods section.

**B** The authors introduce very briefly, why the analysed parameters matter. Apart from lines 14-19 which sketch a few possible impacts of Northeast Atlantic storms, there is nothing. In particular, they provide no reasoning why wind direction is interesting besides wind speed/storm activity alone. A number of reasons are very clear to me but readers not familiar with the specific of potential storm impacts will have no clue why wind direction is relevant beyond academic exercise.

**Response:** We thank the reviewer for this observation. We revised the introduction to include a paragraph that motivates the relevance of wind direction analysis. In particular, we emphasized its importance for understanding the pathways of storm systems, storm surge impacts on coastal areas, directional wind stress on infrastructure, and compound flooding risks due to onshore winds. Furthermore, wave height and direction are dependent on the wind direction (via the wind fetch), and play a big role in determining coastal impacts such as erosion. This will help clarify the added value of this analysis for readers unfamiliar with storm impact mechanisms.

**C** I am a little worried regarding the compilation of the multi-model ensemble. The authors themselves stress a problem related to their procedure of selecting one ensemble member from each model which leads to thos models with just one member always remaining part of the multi-model ensemble. This is a major shortcoming of this study. I am grateful that the authors have the courage to outline this shortcoming themselves. This is a great example of good scientific practice and they present an approach to asses the potential influence by compiling a second multi-model ensemble, bootstrapping only from thos models with 5 or more members. Anyway, I am worried for one more reason: If I understand correctly, the authors perform the bootstrapping for those models with more than one member separately for the historical experiment and the individual scenarios. When doing so, a chosen scenario run is probably unrelated to its historical counterpart (at least in most cases). This yields inhomogeneities masking physical meaningful climate signals. In the end, it probably does not matter so much, given that the averaging over the multii-model ensemble is done, smoothing out this inconsistency between the end of the historical period and three separate beginnings of future scenarios. However, I would argue that choosing scenario simulations belonging to the chosen historical parent simulation had been a better solution. If my understanding is correct, I would ask the authors to include this issue in the discussion, too.

**Response:** We greatly appreciate this comment. The reviewer is correct in assuming that the bootstrapping is performed separately for the historical and the scenario runs. The main reason why we did not choose the same runs for historical and scenarios is that not all members continue from the historical to the scenario period. In other words, the number of available runs changes between historical and the different scenarios for certain models, and some scenario runs do not have a clear parent in the historical period and vice versa. We looked into the feasibility of just using those members for which there exists a clear historical-scenario connection, but figured that this would reduce the number of potential models with >5 members too far to be considered a representative multi-model ensemble. We expanded the discussion to

reflect on this issue and also motivate a more consistent ensemble size for historical and scenario runs in future CMIP modelling efforts.

**D** lines 113-115: Choosing the nearest grid point from each model for a given observational site may lead to some distortion in areas of steep orographic gradients, namely Bodoe and Bergen. I understand that the authors use SLP, however, an extrapolation of modelled surface pressure into the ground of the one model (where the closest grid point is inland) may be somewhat different from SLP practically identical to modelled surface pressure of another model (where the closest grid point is flat terrain or even ocean). I do not insist a priori on including a discussion of this matter in the manuscript but I challenge the authors to think about this possible case (or perform some sensitivity test) and provide arguments here, why this should not be relevant for their findings.

**Response:** We thank the reviewer for this thoughtful comment. We added a discussion in the methods section acknowledging that the selection of the nearest model grid point may introduce slight distortions in orographically complex regions such as Bodø and Bergen and that this is a common issue among all studies that use MSLP-based proxies

**E** I am pretty confused by the section title of Sec. 3.2. "Internal variability" refers to temporal fluctuations of various variables inherent to the natural earth system, observed or modelled. Single-model initial condition large ensembles (SMILE) are great tools to distinguish extrenally forced signals from internal variability. However, this is not what you are analysing in the large part of this section. Instead, the main focus of this section are heterogeneous climate change signals for different parts of the storm intensity spectrum as well as depending on wind direction. I would suggest choosing a more suitable title for Sec. 3.2 plus deleting a few sentences in the first paragraph of Sec. 3.2 (see my specific comment below).

**Response:** We appreciate this concern regarding the title of Section 3.2. In accordance to our response to the specific comment below, we reworked the entire first part of this section so that it does not mix up changes in the ensemble mean and internal variability anymore. We also renamed the section to "A SMILE approach with the high-frequency MPI-GE CMIP6".

**Specific comments:**

**L26-28** Actually, the synthesis of Feser et al. is the other way around (increasing north of 60°N, decreasing south of 60°N) in line with a poleward shift of the NH storm track.

**Response:** We thank the reviewer for catching this error. Indeed, the synthesis of Feser et al. concludes that a majority of studies project decreasing storm activity south of 55-60°N and increasing storm activity north of that. We corrected this error.

**L38-39** I don't understand the line of argumentation here. The previous sentence is about low model agreement, hence large model-related uncertainty. This sentence here now seems to present a link to substantial changes in wind extremes when combined with changes in storm track locations. It seems something is missing here in between.

**Response:** We intended to argue that the large model-related uncertainty in both the location/shift of the storm tracks and the cyclone density leads to a very high uncertainty in the future evolution of (extreme) wind speed distributions at certain locations in the North Atlantic sector. We rephrased this paragraph to clarify our line of argumentation.

**L97** Did the authors analyse if discrepancies are introduced when using 3-hourly data for MPI-ESM compared to daily averages for the other models? An average of the eight 3-hourly values is usually in quite good agreement with a respective daily mean. So, it might even haven been an alternative to calculate such a proxy daily mean for MPI-ESM before analysing its simulations consistently with the other models.

**Response:** We appreciate this concern. For the multi-model analysis, we used daily-mean pressure data only, also from the MPI-ESM model. The three-hourly data are used exclusively in the single-model large-ensemble analysis. We see that this is not explained clearly enough in the Methods section and added a

more detailed description to clarify that we use daily values for the multi-model part, and three-hourly data for the MPI-GE single-model analysis. We agree with the reviewer that directly comparing storm activity values calculated from three-hourly data with those calculated from daily values would introduce inconsistencies to a multi-model framework. We still decided to use the higher-frequency output of the MPI-GE for the single-model section as it allows us to look at more extreme wind speeds, which are not present in daily-mean data.

**Sec. 2.2** It is not entirely clear from this section if MPI-GE is also used as part of the CMIP6 multi-model ensemble and hence included in respective analyses or not.

**Response:** We apologize for the unclarity in this regard. The daily output of the MPI-GE (which is the large-ensemble produced with MPI-ESM-LR) is part of the CMIP6 multi-model ensemble. In the multi-model sections, we use daily-mean SLP data from all listed CMIP6 models, including the MPI-GE (MPI-ESM-LR). The high-frequency output of the MPI-GE is analyzed separately. We see that assigning two names to the same model (MPI-GE and MPI-ESM-LR) might be confusing to the reader, and improved our terminology, i.e., using MPI-ESM-LR for the standard model output, and MPI-GE when talking about the separate three-hourly dataset.

**Sec. 2.3** This section lacks the information that an averaging over all ten triangles is performed to yield results for the Northeast Atlantic. This fact is explicitly written only in lines 271-272, that is the discussion.

**Response:** We agree with the reviewer that this important step is missing here. We added a note on that in the respective section.

**L115-117** Sory, I don't understand the procedure for "triangles" that basically fall onto a line. I am lost when you write about "the observation site that is most distant to the corresponding gridpoint". Which gridpoint? It is three observation sites with three associated grid points... Please rephrase (or correct?) your description here.

**Response:** We apologize for the lack of clarity in this section and thank the reviewer for pointing it out. The procedure aims to construct a triangular area for each model that mirrors the geometry of the three observational sites used to estimate storm activity. For each model, we identify the grid points that are geographically closest to the three observational stations. These three grid points define a triangle over which the geostrophic wind is calculated.

However, in some models, the three closest grid points can fall nearly on a straight line (e.g., sharing the same latitude or longitude), which would prevent a meaningful calculation of the wind vector due to an enclosed area of zero. In such cases, we slightly adjust the position of one grid point to form a proper triangle. Specifically, we move the grid point corresponding to the observation site that is geometrically furthest from the initially assigned grid point. This adjustment is limited to a single grid cell in the nearest orthogonal direction to preserve the original geometry as closely as possible while ensuring a valid triangle. We added an updated description to the manuscript to explain this procedure more clearly.

**L125&149**: I would refrain from using the term "multidecadal oscillation" here. For me this term implies some type of natural variability associated with these fluctuations. But this is not possible given that you analyse the ensemble mean (and certain ensemble quantiles) here. These fluctuations must be either by chance (unlikely in these cases) or also result from external forcing common to all individual simulations. If you discuss these fluctuations, you have to provide possible drivers here.

**Response:** We agree that the term "multidecadal oscillation" is inappropriate here as the ensemble mean can only reflect variability that relates to an external forced climate signal. We rewrote the respective sentences to avoid suggesting that the ensemble mean variability hints at an oscillation originating from within the Earth system.

**Fig. 3** I would suggest swapping subfigures vertically in order to present the same order as in previous figures: first, the results from the bootstrapped MME, second, the results from all members.

**Response:** We changed the order of the subfigures to follow the order of "bootstrapped first, all members second". We also adjusted the captions and references in the text accordingly.

**L178-179** "Internal variability" refers to fluctuations inherent to the natural earth system, observed or modelled. Single-model initial condition large ensembles (SMILE) are great tools to distinguish extrenally forced signals from internal variability. However, you cannot diagnose the internal variability from the SMILE's ensemble mean. It seems to me that you are doing that here. I would suggest eliminating everything from line 178 to line 196, replacing that with a better introduction for the direction- and intensity-dependent analyses, and then continue with line 197.

**Response:** We thank the reviewer for bringing this up. We agree that the ensemble mean of the SMILE is not a suitable tool to quantify internal variability. We see the need to disentangle this paragraph into a part on the general evolution of the SMILE ensemble mean across the scenarios (i.e., the forced signal), which we examine to make sure the MPI-GE is a feasible tool representative of the CMIP6 multi-model signal, and the actual analysis on the internal variability (extremes, wind directions) which uses the pooled ensemble data and not just the ensemble mean. We rewrote the first part of this section, clearly stating that we focus on the large ensemble to detect changes in extremes and distributions, and not to estimate internal variability from the ensemble mean.

**L201** It looks to me as if the (positive) frequency changes are rotated clockwise (not counter-clockwise) compared to the historical frequencies.

**Response:** The reviewer is correct that the positive (red) frequency changes are rotated clockwise compared to the historical (gray) frequencies. However, in this sentence we refer to the differences in frequency changes between the MPI-GE (Figure 6b) and the CMIP6-MME (Figure 4b), which show a counterclockwise shift from CMIP6-MME to MPI-GE. We rephrased this sentence to clarify which wind roses we would like the reader to compare.

**L206-208 ff.** Why are the changes of upper percentiles of wind speeds only analysed from MPI-GE. This could have been done from the CMIP6-MME as well or not?

**Response:** The reason we performed the extreme wind percentile analysis only for MPI-GE is the availability of high temporal resolution data (3-hourly), combined with the large ensemble size within a single model, i.e. with consistent model physics, that allows us to examine absolute wind speeds without having to worry about inter-model biases in non-standardized space. Most CMIP6 models only provide daily data, which is less sufficient for robust storm activity estimation at the high end of the distribution, and those with 3-hourly resolution have fewer ensemble members. We clarified this choice in the manuscript.

**L210** Here you write about "significantly weaker". Have you checked statistical significance? If not, refrain from using this term here but as said in my general comment A) you will have to assess statistical significance.

**Response:** We agree that the term "significantly" should not be used without supporting statistical tests. In line with our response to general comment A, we now include significance tests for these results and updated the figures appropriately. We can confirm that the changes for $95^{th}$ percentiles of SW and NW winds are indeed significant at the 0.05-level, while changes in the intensity of W winds are n

**L219-222** Why does only MPI-GE allow for the analysis of the extreme events? You could have done the same with the full CMIP6-MME, or not?

**Response:** Similar to the point above, we focus on MPI-GE for extreme event analysis due to its temporal resolution, ensemble size, and the ability to keep the analysis within one physically consistent single-model framework without introducing inter-model biases. However, we acknowledge that a subset of CMIP6 models could be used for similar analysis if suitable data are available. We added this information to the section.

**L304-305** Rephrase this sentence about the internal variability. This is not what you are looking at.

**Response:** We removed the sentence with "internal variability" here and added that we analyze the single-model MPI-GE to estimate changes in absolute wind speeds.

**Outlook** I would suggest at least one or two outlook sentences... What are possible next steps that could be done based on your findings?

**Response:** We thank the reviewer for this suggestion. We added a short outlook section at the end of the discussion, highlighting potential next steps such as including a seasonal decomposition, applying percentile-based event attribution approaches, and investigating compound coastal hazard risks with hybrid approaches under future storm conditions.

**Technical corrections:**

We implemented the technical corrections suggested by the reviewer.

We, the authors, would like to thank the two reviewers again for their careful reading of our manuscript and for the constructive comments. We hope that our responses and revisions clarified all outstanding points. We also would like to extend our gratitude again to Jadranka Sepic for processing our manuscript as the handling editor.

With kind regards,

Daniel Krieger and Ralf Weisse